

# Mobile water vapor Raman lidar for heavy rain forecasting: system description and validation

Tetsu Sakai[1], Tomohiro Nagai[1], Toshiharu Izumi[2], Satoru Yoshida[1], Yoshinori Shoji[1]

[1]Meteorological Satellite and Observation System Research Department, Meteorological Research Institute, Tsukuba, 305-0052 Ibaraki, Japan
[2]Observation Department, Japan Meteorological Agency, 1-3-4 Otemachi, Chiyoda-ku, 100-8112 Tokyo, Japan

*Correspondence to*: Tetsu Sakai (tetsu@mri-jma.go.jp)

**Abstract.** To improve the lead time and accuracy of predictions of localized heavy rainfall, which can cause extensive damage in urban areas in Japan, we developed a mobile Raman lidar (RL) system for measuring the vertical distribution of the water vapor mixing ratio ($w$) in the lower troposphere. The RL was installed in a small trailer for easy deployment to the upwind side of potential rainfall areas to monitor the inflow of moist air before rainfall events. We describe the lidar system and present validation results obtained by comparing the RL-measured data with collocated radiosonde, Global Navigation Satellite System (GNSS), and high-resolution objective analysis data. The comparison results showed that RL-derived $w$ agreed within 10% with values obtained by radiosonde at altitude ranges between 0.14 and 1.5 km in the daytime and between 0.14 and 5–6 km at night in the absence of low clouds; the vertical resolution of the RL measurements was 75−150 m, their temporal resolution was less than 20 min, and the measurement uncertainty was less than 30%. RL-derived precipitable water vapor values were similar to or slightly lower than those obtained by GNSS at night, when the maximum height of RL measurements exceeded 5 km. The RL-derived $w$ values were at most 1 g/kg (25%) larger than local analysis data. Four months of continuous operation of the RL system demonstrated its utility for monitoring water vapor distributions for heavy rain forecasting.

## 1 Introduction

In recent years, the occurrence frequency of localized heavy rainfall capable of causing extensive damage has been increasing in urban areas of Japan (Japan Meteorological Agency (JMA), 2016). For early prediction of heavy rainfall, a numerical weather prediction (NWP) model is employed along with conventional meteorological observation data. However, the lead time (period of time between the issuance of a forecast and the occurrence of the rainfall) and accuracy of the prediction are limited, in part, because of the coarse temporal and spatial resolutions of water vapor distribution observations. To improve those observations, we developed a mobile Raman lidar (RL) system that can continuously measure the vertical distribution of water vapor in the lower troposphere. The RL can be easily deployed at a site upwind of a potential heavy rainfall area to monitor the vertical water vapor distribution before a rainfall event. The observed data can then be assimilated by the local ensemble transform Kalman filter method (Kunii, 2014) into a nonhydrostatic mesoscale model (Saito et al., 2007) to improve the initial condition of the water vapor field and consequently the rainfall forecast. We discussed with scientists involved in





model development and implementation the required temporal and spatial resolution and accuracy of the measured data for heavy rain forecasting (Table 1). For example, Kato (2014) has reported that the equivalent potential temperature, which is a function of the water vapor concentration at an altitude of 500 m, is an important parameter for forecasting heavy rainfall in the Japanese area because the inflow of moist air, which can cause heavy rain, mainly occurs at around that altitude. Thus, the

measurable altitude range must extend upward to at least that altitude. In addition, the temporal resolution of the data should be better than 30 min, because the assimilation window can be less than 30 min long. In addition, for data assimilation, the measurement uncertainty (observation error) must be specified. Wulfmeyer et al. (2015) discusses in more detail the requirements of measurements used for data assimilation. We developed our mobile RL system to meet these requirements as much as possible. The RL technique is a well-established technique for measuring the water vapor distribution in the

troposphere (e.g. Melfi et al., 1969, Whiteman et al., 1992), and RL systems have been in operation for decades at stations around the world (Turner et al., 2016; Dinoev et al., 2013; Reichardt et al., 2012; Leblanc et al., 2012). Field-deployable systems have also been developed by several institutes (Whiteman et al., 2012; Chazette et al., 2014; Engelmann et al., 2016). Our RL system is a compact mobile system that can be deployed on a standard vehicle and operated unattended for several months by remote control. Here, we describe our mobile lidar system and present validation results obtained by comparing the

RL-measured data with data obtained by other humidity sensors as well as objective analysis data. Section 2 of this paper describes the RL instrumentation and the data analysis method. Section 3 presents the validation results obtained by comparing the RL measurements with collocated radiosonde measurements, GNSS data, and high-resolution objective analysis data provided by the JMA. Section 4 is a summary.

**Table 1.** Lidar data requirements for localized heavy rain forecasting

| System | Field deployable |
|---|---|
| Measured quantity | Water vapor mixing ratio ($w$) |
| Data description | |
| Altitude range | <0.2 km to >2 km |
| Time period | 24-hour, continuous |
| Vertical resolution | |
| | <200 m |
| Temporal resolution | <30 min |
| Uncertainty | |
| | <10% |



## 2 Instrumentation

### 2.1 Transmitter and receiver optics

The RL system employs a Nd:YAG laser (Continuum Surelite EX) operating at 355 nm with pulse energy of 200 mJ and a repetition rate of 10 Hz. The beam diameter is expanded fivefold to a diameter of ~5 cm by a beam expander (CVI, USA), and

5 the beam is emitted vertically into the atmosphere. The light backscattered by atmospheric gases and particles is collected by a custom-made Cassegrain telescope (primary mirror diameter of 0.35 m, focal length 3.1 m; Kyoei Co., Japan). The focal point of the telescope is within the tube to shorten the length of the receiving system. Light baffles placed inside the telescope tube prevent stray light from entering the detectors. The received light is separated into three spectral components, Raman water vapor (407.5 nm), nitrogen (386.7 nm), and elastic (355 nm) backscatter light, with dichroic beam splitters and

10 interference filters (Barr Materion, USA) and shortcut filters (Kenko, Japan) and detected by photomultiplier tubes (PMTs) (R8619, Hamamatsu, Japan). The interference filter angles of the Raman channels are tuned manually to maximize the transmission of the Raman backscatter signal. To avoid signal saturation of the PMTs, we inserted neutral density filters before the PMTs. The signals are acquired with a transient recorder (Licel TR-20-160) operating in analog (12-bit) and photon counting (20 MHz) modes. The data are stored on the hard disk of a personal computer (PC). The RL can be operated remotely

by issuing commands (e.g. turn high voltage of PMTs on/off, start/stop lasing, start/stop data acquisition, and transfer data) to the PC via wireless Internet communication (Table 2, Fig. 1)

Table 2. Specifications of the mobile Raman lidar

| Transmitter: | | | |
|---|---|---|---|
| Laser | Nd:YAG | | |
|   Wavelength (nm) | 355 | | |
|   Pulse energy (mJ) | 220 (maximum) | | |
|   Repetition frequency (Hz) | 10 | | |
|   Beam divergence (mrad) | 0.125 | | |
| **Receiver:** | | | |
|   Telescope type | Cassegrain | | |
|   Diameter of primary mirror (m) | 0.35 | | |
|   Field of view (mrad) | 0.29 | | |
|   Detectors | Photomultiplier tubes | | |
|   Data acquisition | Photon counting/analog | | |
| **Detection specifications:** | Raman water vapor | Raman nitrogen | Elastic |
| Interference Filter | | | |
|   Center wavelength (nm) | 407.65 | 386.65 | 354.63 |
|   Bandwidth (nm) | 0.25 | 0.34 | 0.6 |
|   Peak transmission (%) | 74 | 45 | 43 |
|   Rejection at 355 nm | $<10^{-13}$ | $<10^{-7}$ | – |





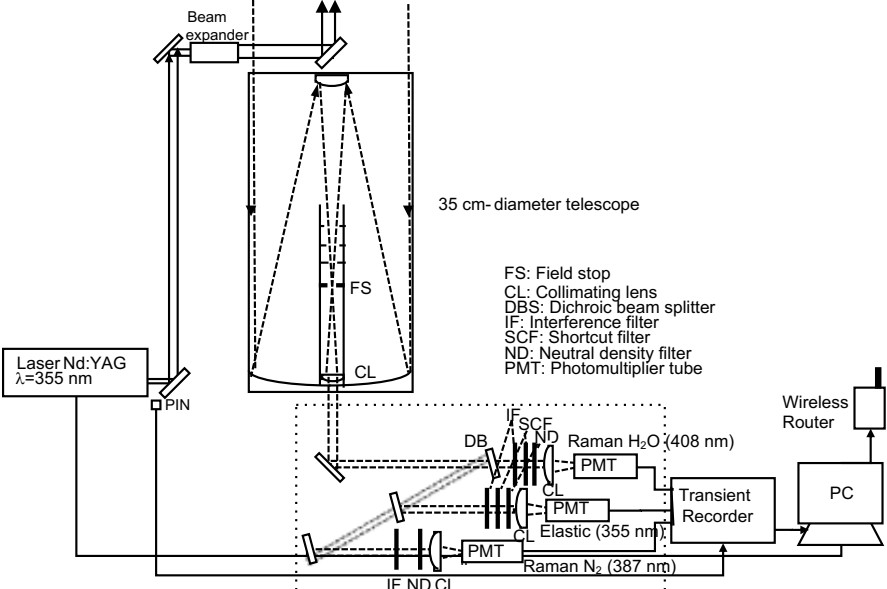

**Figure 1.** Schematic diagram of the mobile Raman lidar system.

**2.2 Trailer**

The RL system is enclosed in a container with outside dimensions of 1.7 m by 4.2 m by 2.1 m high (Figs. 2 and 3). The total

5   weight, including the lidar system and the trailer, is approximately 800 kg. The trailer can be towed behind any standard-sized
vehicle; therefore, anyone who holds a basic-class driver's license can tow it in Japan. The temperature inside the trailer is
maintained to 22–32 ℃ by an air conditioner. A fused silica window (47 cm × 42 cm × 1 cm thick) with an antireflection
coating installed at a tilt angle of 10° above the receiving telescope enables the RL to be operated regardless of the weather.
To prevent direct sunlight from entering the telescope, a chimney-type light baffle with a height of 2 m is mounted on top of

10  the trailer. The system requires a single-phase, three-wire type 100/200V power supply with a maximum current of 10A (5–7
A during normal operation).





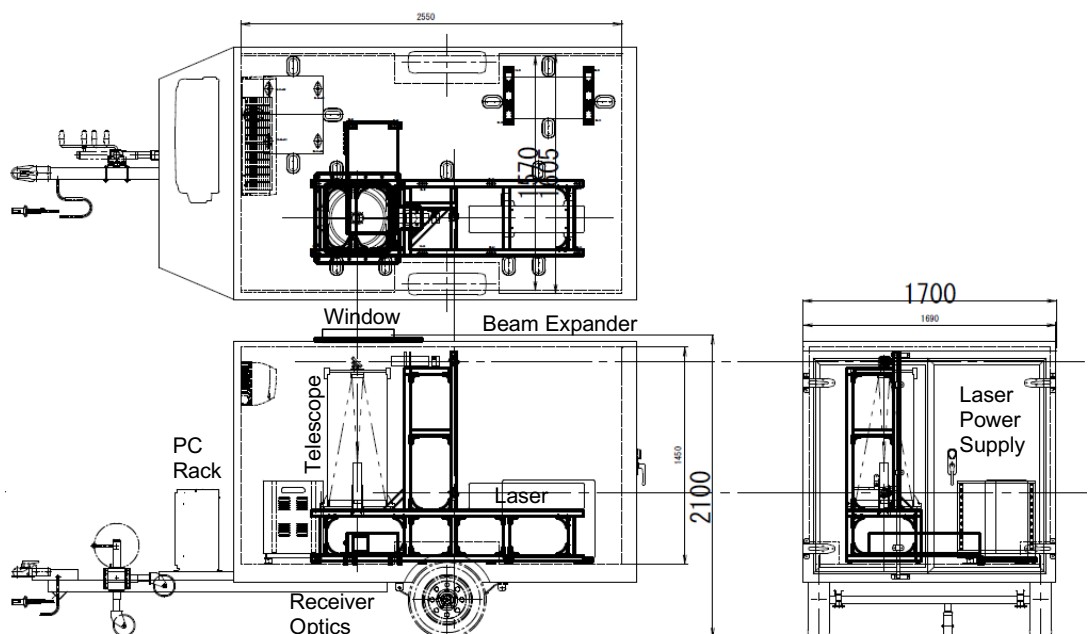

Figure 2. Layout of the mobile Raman lidar system in its trailer. Dimensions are in millimeters.

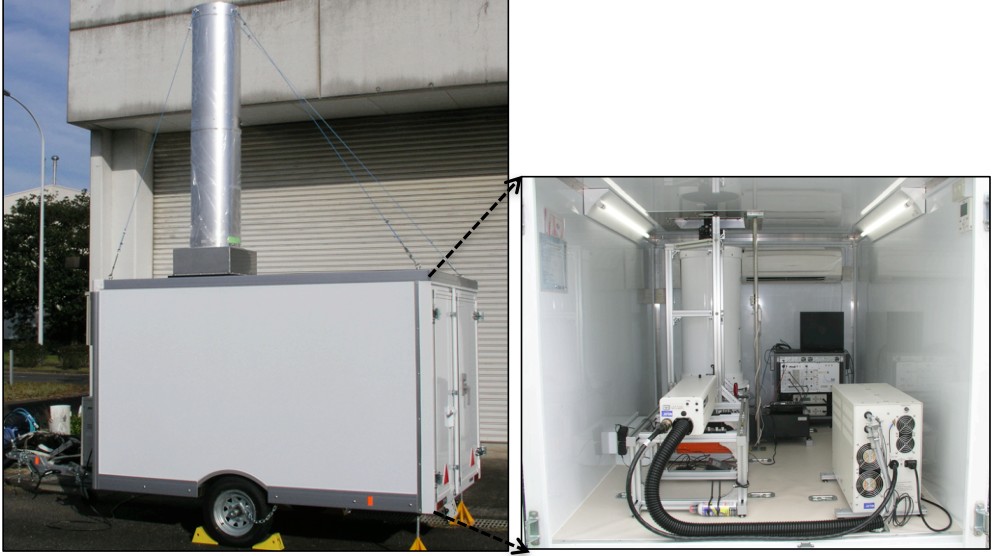





**Figure 3.** Photographs of the mobile RL trailer (left) and its interior (right).

**2.3 Data analysis**

The water vapor mixing ratio ($w$) is obtained from the observed Raman backscatter signal of water vapor and nitrogen as follows:

$$w(z) = K \frac{O_{H_2O}(z)}{O_{N_2}(z)} \frac{P_{H_2O}(z)}{P_{N_2}(z)} \Delta T(z_0, z),$$

with

$$\Delta T = \frac{e^{\int_{z_0}^{z} [\alpha_{H_2O}^m(z') + \alpha_{H_2O}^p(z')] dz'}}{e^{\int_{z_0}^{z} [\alpha_{N_2}^m(z') + \alpha_{N_2}^p(z')] dz'}},$$

(1)

where $K$ is the calibration coefficient of the water vapor mixing ratio, $O_X(z)$ is the beam overlap function of the receiver's channel, and $P_X(z)$ is the noise-subtracted Raman backscatter signal of molecular species $X$ ($H_2O$ or $N_2$) at height $z$ from the lidar at $z_0$, $\Delta T$ is the transmission ratio of the Ramen signals between the lidar at $z_0$ and $z$, and $\alpha_X^m$ and $\alpha_X^p$ are the molecular and particle extinction coefficients of $X$ at the wavelength of the Raman scattering. The value of $K$ was obtained by comparing the uncalibrated RL-derived value of $w$ (i.e. $w$ computed assuming $K = 1$ in Eq. 1) with $w$ obtained with a radiosonde launched 80 m northeast of the RL at 20:30 LST by a weighted least squares method (Sakai et al., 2007) between altitudes of 1 and 5 km and taking the average over the measurement period. See Sect. 2.4 for the values of $K$ obtained in this manner and their temporal variation. In this system, the ratio of the beam overlap functions ($\frac{O_{H_2O}(z)}{O_{N_2}(z)}$) is 1 above an altitude of 0.5 km, and below that altitude it deviates slightly from 1; these values were determined by comparing the RL-derived value of $w$ without overlap correction (i.e. $w$ obtained by assuming $\frac{O_{H_2O}(z)}{O_{N_2}(z)} = 1$ in Eq. 1) with $w$ obtained by radiosonde measurements (see Sect. 2.5). To determine $\Delta T$, we calculated $\alpha_X^m$ using molecular extinction cross section (Bucholz, 1995) atmospheric density obtained from the radiosonde measurement made closest to the RL measurement period; we did not take the differential aerosol extinction for the two Raman wavelengths into account because it is usually less than 5% below the altitude of 7 km (i.e. $\Delta T$ ranges from 1 to 0.95 from the lidar position to 7 km) under normal aerosol loading conditions (Whiteman et al., 1992). The temporal and vertical resolutions of the raw data were 1 min and 7.5 m, respectively. To reduce the statistical uncertainty of the derived $w$, we averaged the raw data over 20 min and reduced the vertical resolution to 75 m below 1 km altitude and 150 m above that. The measurement uncertainty of $w$ was estimated from the photon counts by assuming Poisson statistics (e.g. Whiteman, 2003) and the uncertainty of the calibration coefficient as follows:

$$\delta w(z) = \left[ \left( \frac{\delta K}{K} \right)^2 + \left( \frac{\delta P_{H_2O}(z)}{P_{H_2O}(z)} \right)^2 + \left( \frac{\delta P_{N_2}(z)}{P_{N_2}(z)} \right)^2 \right]^{\frac{1}{2}},$$

where

$$\delta P_X = \left( P_{X,signal} + 2P_{X,noise} \right)^{\frac{1}{2}}.$$

(2)





The signal ($P_{X, signal}$) was obtained from the total backscatter signal by subtracting the background noise ($P_{X, noise}$), which was computed by taking the average of the total signal between the altitudes of 80 and 120 km, where atmospheric backscattering was expected to be negligible. The uncertainty of the calibration coefficient ($\delta K$) was estimated as the standard deviation of $K$, which was obtained from the comparison of uncalibrated RL-derive data with the radiosonde data for the measurement period.

As quality control (QC) of the derived data, we excluded data with uncertainty larger than 30% or $w > 30$ g/kg.

**2.4 Calibration coefficient of the water vapor mixing ratio**

To obtain the absolute value of $w$ from the lidar signals, the calibration coefficient $K$ of Eq. (1) was first determined as described in Sect. 2.3. However, temporal change in $K$ is a critical problem for long-term operation of the system, because if the temporal variation is large, $K$ must be obtained frequently during the measurement period. We investigated this problem by examining

the temporal variation in $K$ values obtained by comparing uncalibrated RL-derived $w$ with collocated radiosonde measurements obtained daily at 20:30 LST from August to December 2016 (Fig. 4). Radiosondes (RS-11G, Meisei Electric. Co., Japan) were launched twice daily (8:30 and 20:30 LST) from an aerological observatory located 80 m northeast of the RL, and, according to the manufacturer, the measurement uncertainty of relative humidity by the RS-11G radiosonde is 5% in the lower troposphere and 7% in the upper troposphere (http://www.meisei.co.jp/english/products/RS-11G_E.pdf). During the test

period, the RL system was operated nearly continuously at the Meteorological Research Institute in Tsukuba, except for short interruptions for flash lamp replacement (31 August), power outages (18 August and 23 October), and trailer inspection (31 October to 6 November). We calculated $K$ only for the nighttime (20:30 LST) data because at night the RL measurement uncertainty was small between altitudes of 1 and 5 km (see Sect. 3.1). After 12 August, the value of $K$ was nearly constant during the test period: mean ± standard deviation = 52.4 ± 2.1 (Fig. 4). Unfortunately, the reason for the abrupt change in $K$

on 11 August from 57.4 ± 1.5 is unknown because we did not make any changes to the instrument at that time. Nevertheless, given the uncertainty of $K$ (4% in this case), we may say that the RL can be operated for at least 4 months without calibration. We also examined the value of $K$ before the system was moved from Tsukuba to the Tokyo Bay area (110 km or 70 km from Tsukuba) with that obtained after the move, from 15 June to 9 November 2017 (not shown). Before the system was moved, $K$ was 46.9 ± 1.8, and afterward it was 43.1 ± 2.3, a change of 8.6% (we note also that after the telescope focus was readjusted

in January 2017, the value of $K$ changed from what it had been in 2016). These results indicate that the calibration coefficient should be determined before and after deployment of the system, and the average and standard deviation of those values should be used for $K$ and $\delta K$.



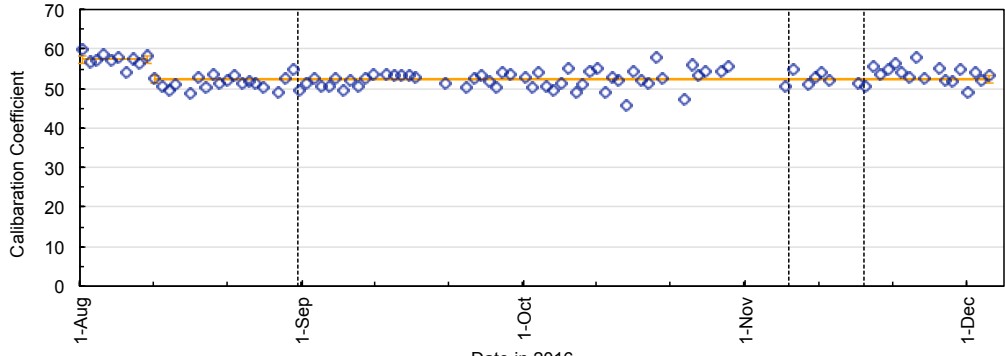

**Figure 4.** Temporal variation of the calibration coefficient of the water vapor mixing ratio ($K$) for the mobile RL obtained by comparison with collocated radiosonde measurements at 20:30 LST from August to December 2016. The horizontal orange lines show the averages before and after 12 August. The vertical dotted lines indicate dates on which the optical axis was

adjusted.

**2.5 Beam overlap correction for the Raman channels**

Values of $w$ calculated from the RL signals for altitudes below 0.5 km were systematically lower than values obtained with the radiosonde when it was assumed that the beam overlap functions for the Raman water vapor and nitrogen channels were equal (i.e. $\frac{O_{H_2O}(z)}{O_{N_2}(z)} = 1$). When we compared the vertical distribution of the ratio of $w$ obtained by radiosonde to that obtained

by the RL without beam overlap correction (Fig. 5), we found considerable variation among individual profiles, but the average value of the ratio increased from 1 to 1.1 with a decrease of altitude from 0.7 to 0.1 km. Possible reasons for the difference in the overlap functions of the two Raman channels at low altitude are the difference in the optical paths (Fig. 1) and the spatial inhomogeneity of PMT sensitivity (Simeonov et al., 1999; Hamamatsu Photonics, 2017). To correct for the difference, we derived the ratio of beam overlap functions by comparing $w$ obtained with the RL under the assumption of $\frac{O_{H_2O}(z)}{O_{N_2}(z)} = 1$ with $w$

obtained by radiosonde. Then, we calculated the mean vertical profile of the ratios and fitted a quadric curve to the profile for use in Eq. (1) to calculate $w$. The magnitude of the correction increased from 1% at 0.5 km altitude to 8% at 0.1 km.





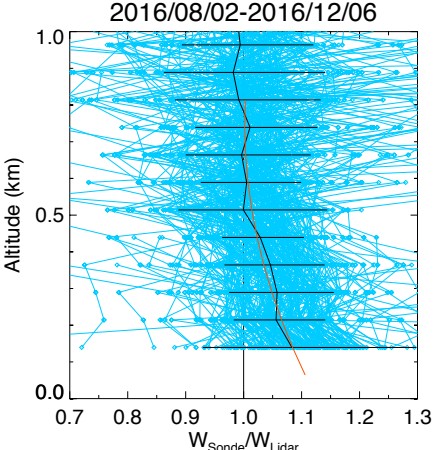

Figure 5. Vertical distribution of the ratio of $w$ obtained by radiosonde ($w_{Sonde}$) to $w$ obtained with the RL system without beam overlap correction ($w_{Lidar}$) from 2 August to 6 December 2016. The individual profiles are shown by the thin blue lines with diamonds. The solid black line and the error bars are averages and standard deviations over 75 m height interval. A quadric

curve (orange line) was fitted to the averaged values.

## 3 Validation results

Measurements for validation of the RL system measurements were made on 120 days, from 2 August to 6 December 2016, over Tsukuba, Japan (36.06ºN, 140.12ºE). We validated RL-derived $w$ values (described in Sect. 2.3) by comparing them with radiosonde, GNSS, and high-resolution local analysis (LA) data. A GNSS receiver 80 m west of the RL observed the

carrier phase transmitted by GNSS satellites and estimated the precipitable water vapor (PWV) with a temporal resolution of 5 min during the validation period. The PWV value represents the vertically integrated water vapor content averaged over a horizontal distance of approximately 20 km around the antenna. See Shoji et al. (2004) for more details of the derivation method. The LA consists of hourly meteorological data with a horizontal resolution of 2 km over Japan provided by the JMA. These data are obtained by a three-dimensional variational (3D-Var) data assimilation technique from hourly

observation data from multiple sources, including surface measurements, satellites, and GNSS-derived PWV data. LA data provide initial conditions to local-scale NWP models used for 9-hour forecasts for aviation, weather warnings and advisories, and very short-range precipitation in and around Japan, provided every hour. The vertical resolution of the LA data is 45– 868 m with 48 layers. See JMA (2016) for more details about the LA data.




### 3.1 Comparison with radiosonde measurements

### 3.1.1. Vertical distribution

We compared the vertical distribution of $w$ obtained with the RL with $w$ obtained by radiosondes launched at 8:30 and 20:30 LST on 1 September 2016 over Tsukuba (Fig. 6). The ascent speed of the radiosondes was 5–6 m/s, so they reached a height

of about 7 km after 20 min. The RL data were accumulated over the 20 min following the radiosonde launch. The vertical resolution is reduced to 75 m below an altitude of 1 km and to 150 m above that to increase the signal-to-noise ratio (SNR) of the Raman backscatter signals. The values of $w$ obtained with the RL agreed well for the altitude range of 0.14–1.7 km with $w$ obtained by radiosonde during 08:30–08:50 LST (Fig. 6a), and they agreed well for altitudes up to 6.2 km with radiosonde measurements made during 20:30–20:50 LST (Fig. 6b). Mean differences were 0.8 g/kg (7%) for the 08:30 LST radiosonde

launch and 0.7 g/kg (15%) for the 20:30 LST launch. The maximum height of RL measurements with an uncertainty of less than 30% was only 1.5 km in the daytime, because solar light reduces the SNR of the Raman backscatter signals; for example, at 08:30 LST on 1 September 2016, the solar zenith angle was 50° (Fig. 6a).

The altitude–time cross section of $w$ obtained with the RL on 1 September 2016 (Fig. 7) showed considerable diurnal moisture variation below an altitude of 3 km. The top height of a moist region ($w > 12$ g/kg) present below an altitude of 1 km

during 00–03 LST increased to above 2 km as the sun rose during 03–06 LST. At midday, the top height of the moist region was probably above 1.5 km (although it cannot be seen because of the low SNR ratio in strong sunlight). After sunset, it remained at an altitude of 2.5 km, which probably corresponded to the top of a residual layer. The top of another moist region with $w$ of 15 g/kg that emerged below an altitude of 1 km after 18 LST undulated with a vertical amplitude of a few hundred meters and a period of ~3 h. This result demonstrates the utility of the RL system for monitoring the diurnal variation of water

vapor in the lower troposphere, which is not captured by routine radiosonde measurements.

To test the long-term stability of the mobile RL system, we operated it for four months, from 2 August to 6 December 2016. After QC of the RL data, the maximum measurement height was mostly ~1 km during the day throughout the measurement period, whereas at night when low, thick clouds were absent, it decreased from 6 km to 2.5 km over the measurement period (Fig. 8). We attribute this nighttime decrease to 1) a drop by almost half (~40%) in the power of the laser

transmitter during its continuous operation for three months, which caused the SNR of the signals to decrease, and 2) decreases in the water vapor concentration from summer to winter in the lower troposphere, which caused a decrease in the strength of Raman backscatter water vapor signals.

In general, vertical distributions of $w$ obtained with the RL system agreed well with radiosonde measurements, but the RL- and radiosonde-derived values sometimes differed considerably from LA data for the same dates (e.g. between 2.5 and

3.5 km at 20:30 LST on 9 August, between 1.5 and 2.5 km at 20:30 LST on 16 September, and between 0.5 and 1.2 km at 20:30 LST on 2 December 2016) (Fig. 9). These results suggest that the assimilation of RL-derived data can improve the initial conditions of the water vapor distribution in NWP models.





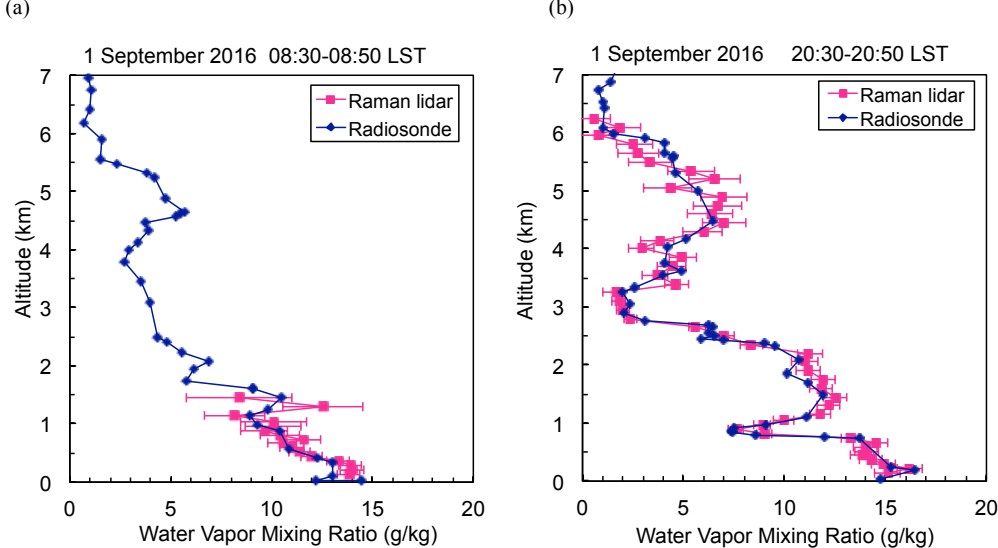

**Figure 6.** Vertical distributions of the water vapor mixing ratio obtained with the mobile RL (magenta), radiosonde (dark blue) on 1 September 2016 over Tsukuba. The measurement periods for the RL were (a) 08:30–08:50 and (b) 20:30–20:50 LST, and the radiosondes were launched at (a) 8:30 LST and (b) 20:30 LST. RL data with uncertainty of less than 30% are plotted.

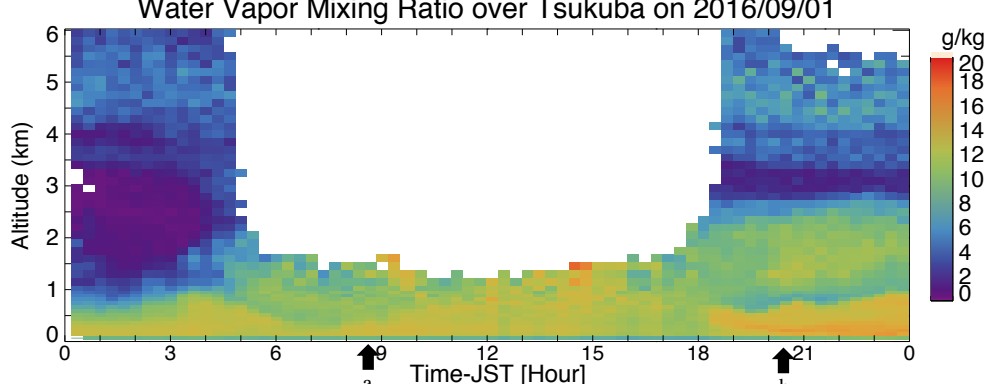





**Figure 7.** Altitude–time cross section of water vapor mixing ratios obtained with the mobile RL on 1 September 2016. Data with uncertainty of less than 30% are plotted. Arrows at the bottom show the start of the measurement periods for the data shown in Fig. 6.

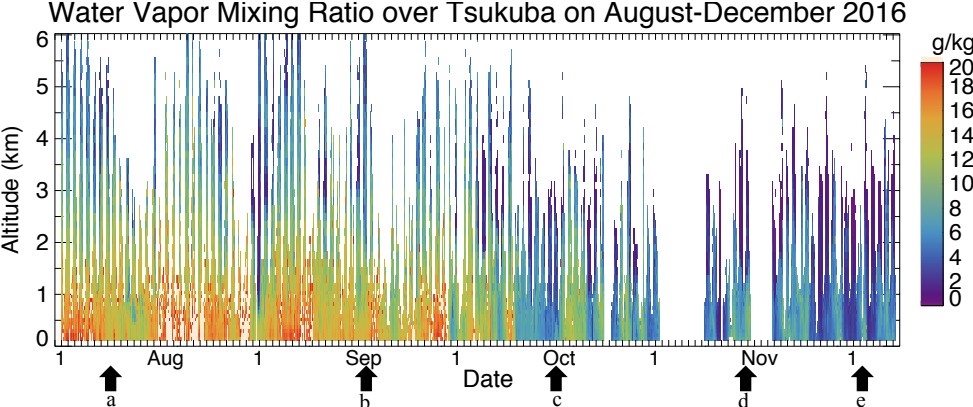

5   Figure 8. Altitude–time cross section of water vapor mixing ratios obtained with the mobile RL from 2 August to 6 December 2016. Data with uncertainty of less than 30% are plotted. Arrows at the bottom show the dates for which vertical profiles are shown in Fig. 9.

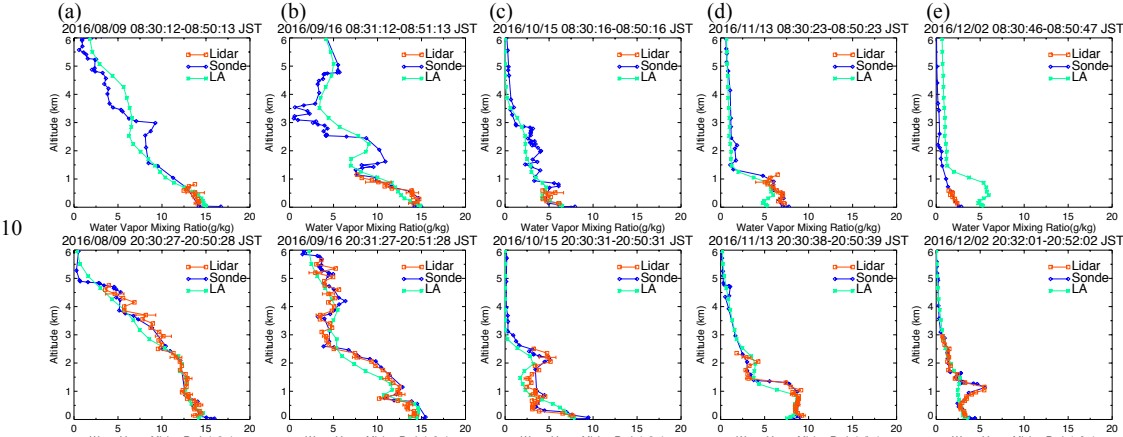

**Figure 9.** Vertical distributions of water vapor mixing ratios obtained with the mobile RL (orange) and radiosondes (blue) compared with local analysis data (green) for 08:30 LST (upper panel) and 20:30 LST (bottom panels) on (a) 9 August, (b) 16 September, (c) 15 October, (d) 13 November, and (e) 2 December 2016.




### 3.1.2 Scatter plot comparison

After the data were screened for QC, we compared $w$ values obtained with the RL and by radiosonde from 2 August to 6
December 2016 in 110 vertical profiles for 20:30 LST and 113 for 08:30 LST (Fig. 10). For this comparison, the radiosonde
data were linearly interpolated to the heights of the RL data. Note that the maximum altitude of the comparison for 08:30 LST

(1.9 km) was lower than that for 20:30 LST (6.85 km) because, owing to their large uncertainty, daytime data at higher altitudes
were excluded by the QC screening. The RL-derived $w$ ($w_{Lidar}$) values agreed with the radiosonde-derived values ($w_{Sonde}$) over
the range from 0 to 20 g/kg (Fig. 10). A geometric mean regression analysis conducted by assuming that $w_{Sonde} = $ slope $\times w_{Lidar}$
+ bias yielded a slope of 0.994 and an intercept of –0.002 for the 20:30 LST (Fig. 10a) and a slope of 1.052 and an intercept
of –0.005 g/kg for 08:30 LST (Fig. 10b). To examine the dependence of the difference in $w$ ($w_{Lidar} - w_{Sonde}$) on the magnitude

of $w_{Sonde}$, we plotted ($w_{Lidar} - w_{Sonde}$) as a function of $w_{Sonde}$, as well as the means and standard deviations of ($w_{Lidar} - w_{sonde}$), at
intervals of 2.5 g/kg (Figs. 10c and 10d). As a result, we found no significant bias in the difference–$w_{sonde}$ relationship for
$w_{Sonde}$ ranging from less than 20 g/kg at night to less than 15 g/kg in the daytime (i.e., mean differences were smaller than 0.3
g/kg). In contrast, we found positive biases for larger $w_{Sonde}$ value ranges; the bias was 1.7 g/kg at 08:30 LST for $w$ ranging
from 17.5 to 20 g/kg. A possible reason for the daytime bias at high values of $w_{Sonde}$ is that high solar background radiation

generated spurious noise spikes and high photon counts in Raman water vapor signals above an altitude of 1 km that were not
rejected by QC. We are investigating the method to reject such data by QC, although they have small impacts on the water
vapor fields analyzed from the data assimilation because their measurement errors are large.

(a) (b)

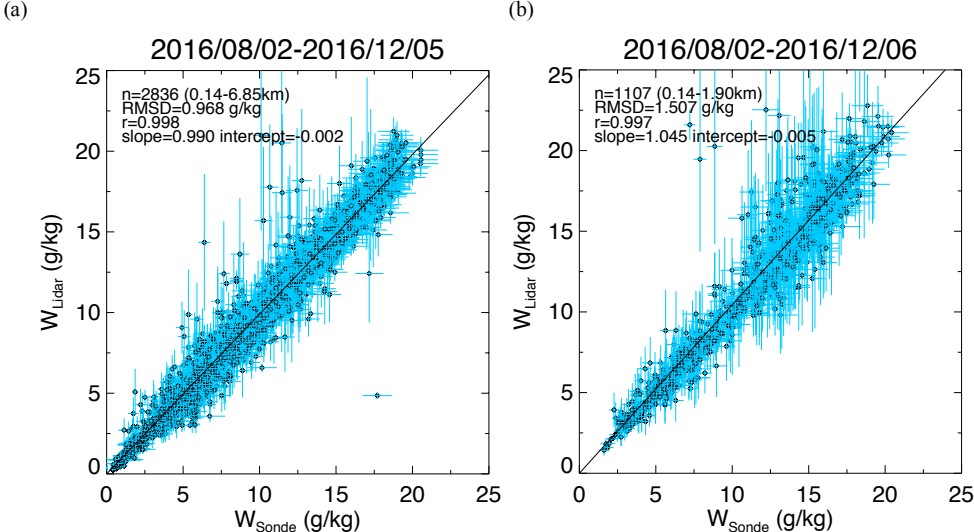



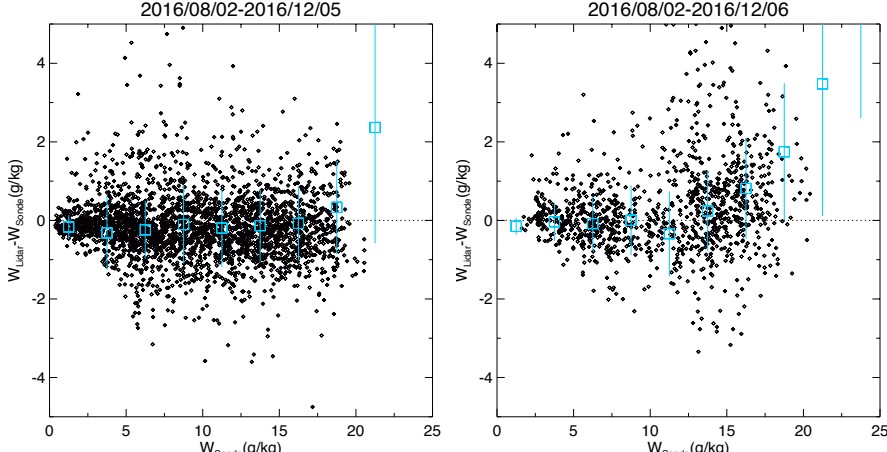

**Figure 10.** (Top panels) Scatter plots of $w$ obtained with the RL ($w_{Lidar}$) versus $w$ obtained with radiosondes ($w_{Sonde}$) at (a) 20:30 LST and (b) 08:30 LST from 2 August to 6 December 2016. (Bottom panels) Scatter plots of the difference ($w_{Lidar} - w_{Sonde}$) as a function of $w_{Sonde}$ at (c) 20:30 LST and (d) 08:30 LST. Blue symbols show the means, and the blue lines show the

5    standard deviations of the difference at intervals of 2.5 g/kg. Data points with an RL measurement uncertainty of less than 30% are plotted.

### 3.1.3 Vertical distribution comparison

To study the height dependence of the difference ($w_{Lidar} - w_{Sonde}$), we examined the vertical variation of the mean difference at intervals of 500 m (Fig. 11). The mean difference was less than 1 g/kg (10%) below an altitude of 6 km at night and below 1

10    km in the daytime. Above these altitudes, the RL values were higher than the radiosonde-derived values. Possible reasons for the larger differences at higher altitudes are 1) the small number of data points in those regions (Fig. 11d), which caused the statistical significance to be low, 2) the difference in the air parcel measured by the two instruments, because as they ascended the radiosondes were sometimes blown several kilometers or more from the RL position by horizontal winds, particularly above an altitude of 6 km at night, and 3) the generation of spurious Raman signals above 1 km by high solar background

15    radiation in the daytime (see Sect. 3.1.2).





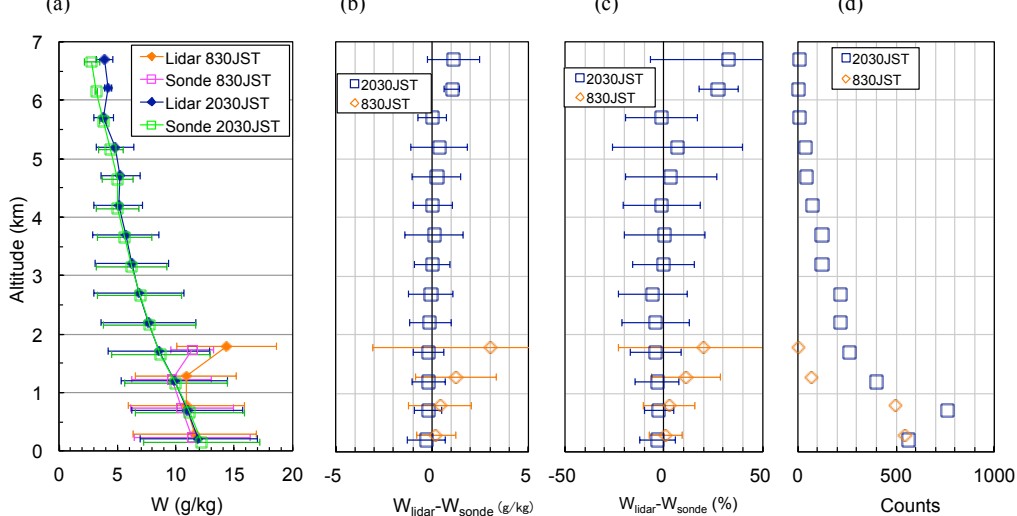

**Figure 11.** Vertical variations of (a) mean $w_{Lidar}$ values (diamonds) and $w_{Sonde}$ (open squares) values at intervals of 500 m for 20:30 LST and 08:30 LST from 2 August to 6 December 2016, and their (b) absolute and (c) relative differences. Symbols and

error bars in (a)–(c) show means and standard deviations. (d) The number of data points at each altitude.

**3.2 Comparison with GNSS PWV data**

To validate the RL measurement data for times when coincident radiosonde data were unavailable, we compared the RL-derived PWV with PWV values obtained from GNSS data. To obtain PWV from the RL data, we computed the vertical profile of the water vapor density from RL-derived $w$ and atmospheric density obtained by the radiosonde closest in time to the RL

measurement period, and vertically integrated the water vapor density from an altitude of 0.14 km to the maximum height with a measurement uncertainty of less than 30%. Below 0.14 km, we assumed that the value of $w$ was the same as that at 0.14 km. Then we compared the temporal variations of PWV obtained with the RL with those obtained from GNSS data from August to December 2016 (Fig. 12). So that this comparison would be meaningful, we excluded RL data obtained when the maximum measurement height was lower than 5 km; as a result, mostly nighttime lidar values obtained when low, thick clouds were

absent were used in the comparison. The temporal resolution of the GNSS data was reduced by averaging from 5 min (original GNSS resolution) to 20 min to match the resolution of the RL data.

The temporal variation of RL-derived PWV was similar to that of the GNSS-derived PWV (Fig. 12). In summer (August–September), when a moist air mass from the Pacific Ocean covered the observation area, the PWV values were mostly higher than 30 mm. In autumn and winter (October–December), when a dry air mass from the Asian continent prevailed, the PWV

values were mostly lower than 20 mm. We note that the number of available lidar PWV data was smaller in autumn and winter




than in summer because the decrease in the laser power as mentioned before (Sect. 3.1.1) and because in autumn and winter the Raman backscatter signal tends to be weak by the low water vapor concentration in the middle troposphere. The regression analysis of PWV derived from RL data against GNSS-derived PWV showed a strong positive correlation (correlation coefficient 0.990; Fig. 13a) between them, but many of the RL-derived PWV values were lower, most by up to 5 mm, than the

GNSS-derived values (Fig. 13b). The most plausible reason for the lower RL-derived PWV values is that the RL did not always measure the entire water vapor column. In addition, both positive and negative differences could be caused by the measured air masses being different (see Sect. 3). The difference in PWV would be large if large horizontal inhomogeneity of the water vapor concentration existed in the observation area. Shoji et al. (2015) utilized the slant path delay of the GNSS signal to estimate the horizontal inhomogeneity of water vapor on a scale of several kilometers around the measurement site. The use

of a technique that combines RL and GNSS observations for monitoring the vertical and horizontal distributions of water vapor holds promise, and the development of such a technique is our future task.

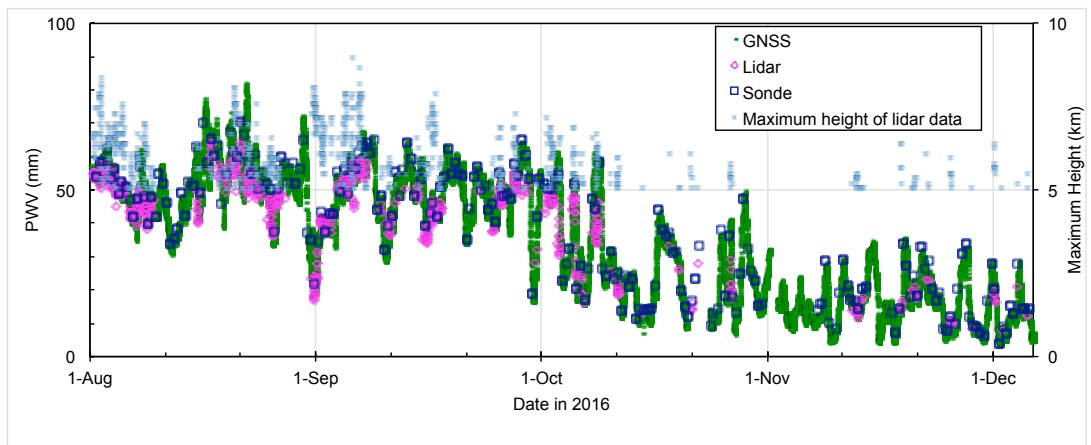

**Figure 12.** Temporal variations of PWV obtained with lidar (magenta diamonds), GNSS (green dots), and radiosonde (blue

squares) from 2 August to 6 December 2016 over Tsukuba. Data with measurement uncertainties of less than 10% that were obtained when the maximum RL measurement height exceeded 5 km (light blue asterisks) are plotted.





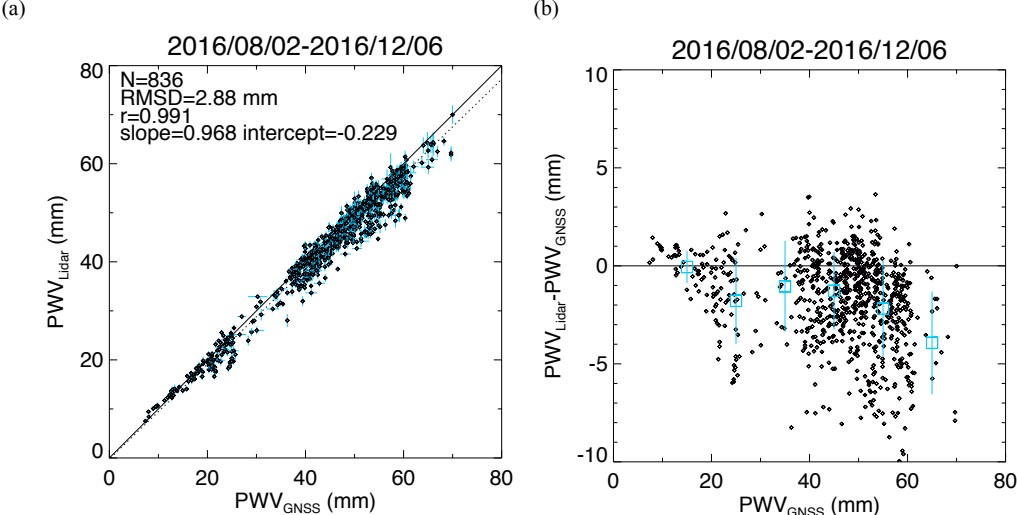

**Figure 13.** Scatter plots (a) of PWV obtained with the mobile RL system against PWV obtained from GNSS data from 2 August to 6 December 2016 and (b) their difference ($PWV_{Lidar} - PWV_{GNSS}$) versus $PWV_{GNSS}$. In (b), the open squares and vertical lines show the means and standard deviations of the difference at intervals of 10 mm.

**3.3 Comparison with local analysis data**

**3.3.1 Scatter plot comparison**

We compared hourly RL values of $w$ with LA data because the primary purpose of our RL measurement was to improve the initial condition of the water vapor field of the NWP model (Fig. 14). For this comparison, the RL data were linearly interpolated to the heights of the LA data. The result revealed that the root mean square difference (RMSD) (1.367 g/kg) was larger than that obtained when we compared RL values with nighttime radiosonde values (0.968 g/kg; Fig. 10a). Moreover, the RL-derived $w$ values were consistently higher, by 0.2–0.8 g/kg (1–11%), than those derived by LA for $w$ in the range of 0–22.5 g/kg (Fig. 14b). We also compared LA data with the radiosonde data for the same period (not shown) and found that the mean LA data at intervals of 2.5 g/kg differed from the radiosonde data by –0.2 to 0.9 g/kg (3–11%). We infer that the LA data used in this comparison had a negative bias because the accuracy of the radiosonde relative humidity measurements was 5–7%.





### 3.3.2 Vertical distributions

Our comparison of vertical variations in $w$ obtained with the RL system with $w$ derived from the LA showed that the RL values were higher by up to 1.1 g/kg (25%) over the entire altitude range (Fig. 15). In addition, the magnitude of the difference ($w_{Lidar} - w_{LA}$) was larger than the difference with radiosonde values ($w_{Lidar} - w_{Sonde}$) (Fig. 11). This result suggests that the assimilation of RL data has the potential to improve the initial conditions provided to the NWP model.

(a)                                                                 (b)

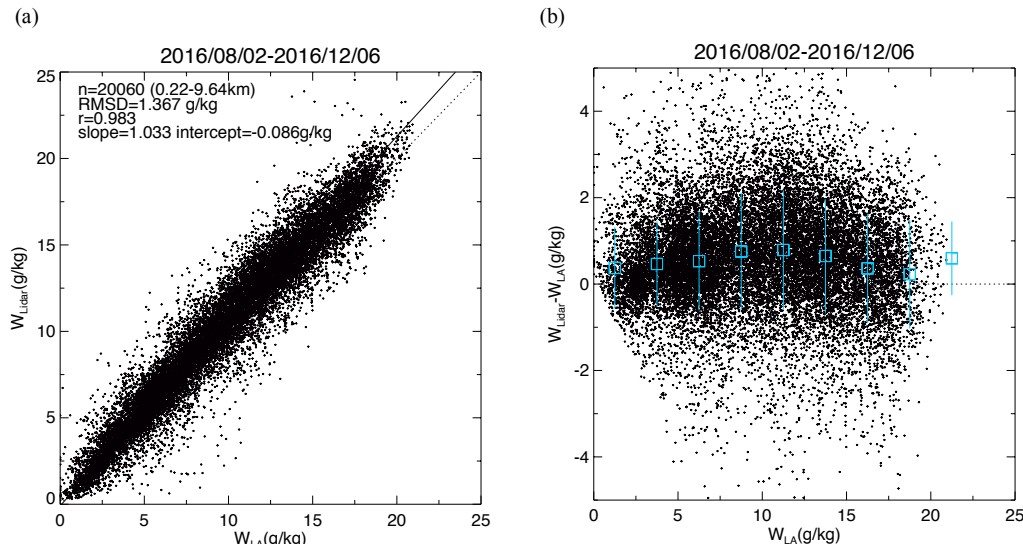

**Figure 14.** Scatter plots of (a) $w$ obtained with the RL ($w_{Lidar}$) versus $w$ obtained from the local analysis ($w_{LA}$) and (b) their difference ($w_{Lidar} - w_{LA}$) as a function of $w_{LA}$ from 2 August to 6 December 2016. In (b), the blue open squares and vertical lines show means and standard deviations of the difference at intervals of 2.5 g/kg.





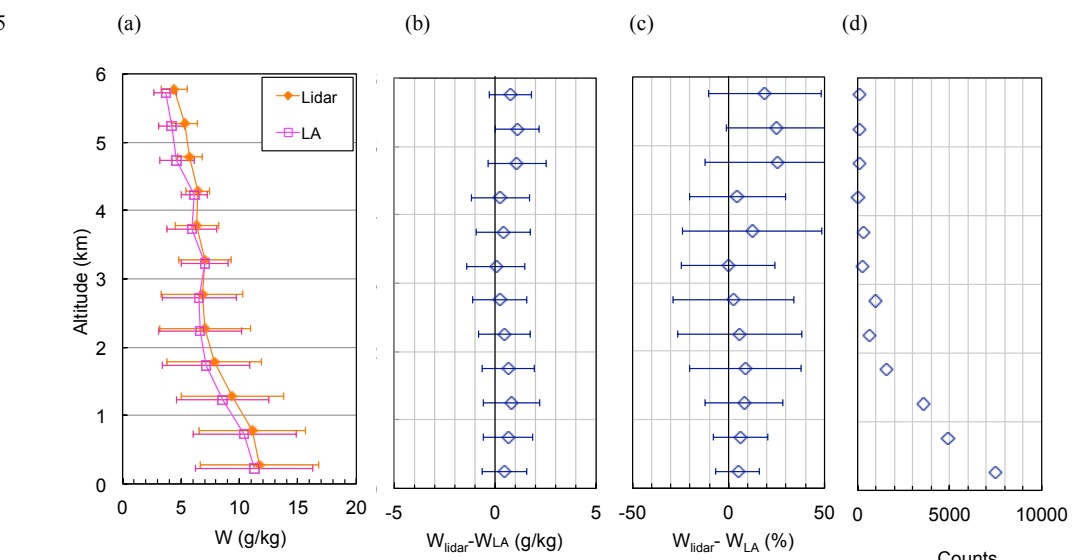

**Figure 15.** Vertical variations of (a) mean values and standard deviations of $w$ obtained with the RL ($w_{\text{Lidar}}$) and from the local analysis ($w_{\text{LA}}$) at 500-m intervals and their (b) absolute and (c) relative differences from 2 August to 6 December 2016. Symbols and error bars in (b) and (c) show the means and standard deviations of the difference. (d) The number of data points at each altitude.

### 3.4 Summary of the validation results

Table 3 summarizes the results of our comparisons of water vapor measurements obtained by the RL and other instruments or local analyses. The correlation was highest and the RMSD was smallest when RL-derived $w$ was compared with $w$ obtained by radiosonde at night. This result was probably because 1) the RL system was calibrated by using radiosonde data, 2) the instruments measured the same quantity ($w$), and 3) the measurement performance of the RL was best at night. The agreement with radiosonde data was not as good in the daytime as it was at night because of the measurement uncertainty of $w$ was larger in the daytime, even though the slope and intercept of the regression analysis did not differ significantly between daytime and nighttime measurements. The RL-derived PWVs at night were slightly lower than those derived from GNSS data because of the measurement range limitation of the RL system. The regression analysis of RL-derived $w$ versus LA data showed that the



magnitudes of the deviation of the slope from 1 and the deviation of the intercept from zero were larger than those obtained in the analysis with radiosonde data, and the correlation coefficient was the lowest among the comparisons.

From these results, we can conclude that assimilation of RL-derived $w$ after QC can improve the initial conditions of the NWP model for heavy rain forecasting.

Table 3. Results of water vapor measurements by the mobile RL compared with data obtained by other instruments or from local analyses.

| Data type | Time (LST) | Slope | Intercept (g/kg) | Correlation coefficient | RMSD (g/kg) | No. of data points |
|---|---|---|---|---|---|---|
| Radiosonde | 20:30 | 0.990 | −0.002 | 0.998 | 0.968 | 2836 |
| | 08:30 | 1.045 | −0.005 | 0.997 | 1.507 | 1107 |
| GNSS (PWV) | 0:00–23:00 | 0.968 | −0.229 mm | 0.991 | 2.88 mm | 836 |
| LA | 00:00–23:00 (hourly) | 1.033 | −0.086 | 0.983 | 1.367 | 20060 |

**5 Conclusion**

We developed a mobile Raman lidar system for measuring the vertical distribution of the water vapor mixing ratio $w$ in the lower troposphere to improve the accuracy and lead time of heavy rainfall prediction. The RL can be easily deployed to remote sites and is capable of unattended operation for several months. Our comparison of the RL-derived $w$ values with those obtained with collocated radiosondes showed that they agreed within 10% between altitudes of 0.14 and 5–6 km at night and between altitudes of 0.14 and 1.5 km in the daytime. The calibration coefficient of the RL showed no significant temporal variation

during 4 months of continuous operation in 2016. A small correction for beam overlap was necessary below 0.5 km. The RL-derived precipitable water vapor values obtained at night when low clouds were absent and the maximum heights of the RL measurement exceeded 5 km were slightly lower than those obtained from GNSS data. The fact that the RL-derived $w$ values were at most 1 g/kg (25%) larger than those in the local analysis data suggests that assimilation of the RL data can improve the initial condition of the water vapor distribution in the lower troposphere of the NWP model. At present, we are studying

the impact of using lidar data with the nonhydrostatic mesoscale model for simulating heavy rainfall in the Kanto area in summer 2016 (Yoshida et al., 2017).

The measurement altitude of the current Raman lidar system is limited to 1.5 km in the daytime. Although this limitation might not preclude the use of data from the system for heavy rain forecasting, it would be better to expand the measurement height range because the mixed layer, where the inflow of the large amount of water vapor that causes heavy rain mostly

occurs, can be as high as 2 km. Moreover, humidity in the middle troposphere affects the development of cumulus convections




to the upper troposphere. To detect water vapor in the middle troposphere in the daytime, a diode laser-based differential absorption lidar might be useful because it can continuously measure the water vapor concentration up to an altitude of 3 km both in the daytime and at night (Repasky et al., 2013; Spuler et al., 2015). We are also developing such a system (Pham Le Hoai et al., 2016) to improve the model forecast skill for heavy rainfall in urban areas.

**Acknowledgements**

We used radiosonde data measured by the Japan Meteorological Agency (downloaded from http://www.data.jma.go.jp/obd/stats/etrn/upper/index.php).

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
