# Peer review of "Automated compact mobile Raman lidar for water vapor measurement: instrument description and validation by comparison with radiosonde, GNSS, and high-resolution objective analysis"

_Atmospheric Measurement Techniques, 2018_

## Referee Comment (RC1) · Anonymous Referee #1 · 19 Jun 2018

The article presented by Sakai et al. is within the topics of AMT. It is clear in its approach and presents a very detailed validation of a new H2O Raman lidar. In addition, the opportunities offered by the water vapor lidars are relevant considering the increase of the extreme raining events. It deserves to be published. I have some minor corrections and remarks listed in the following:

Abstract L15. Changed their by the Introduction The requirement for data assimilation is more on the absolute value of the root-mean-square-error, less than 0.4 g/kg in the planetary boundary layer (Weckwerth et al). Biases are more problematic for data assimilation process and may induce large discrepancies.

Section 2.4 The investigation about the variation of K is a very interesting study. From our experience, it may be due to temperature instabilities in the trailer, although in our

case we could not pinpoint whether it was due to PMT gain or filter CWL variations. Maybe the temperature of the air conditioning was set differently during the summer and the fall? The high voltages of the photomultipliers seem to be fixed; some authors vary the PMT gains to adjust for the signal/sky background noise ratio during daytime and nighttime. This means the PMT gain has been optimized for daytime limitations, and that the lidar could be more effective at night. Could you comment? Why was it necessary to adjust the focus? Because of the displacement of the trailer? Is your collimating lens an achromat? If not, it could explain the change of K with the change of focus.

Section 2.5 To improve the calibration process, especially for the overlap correction function in the lower layers, tethered balloon or kite can be used as in Totems and Chazette (2016). We are then certain of the location of the reference measurements, and we can renew it at will. The accuracy on w is then better. Totems, J. and Chazette, P.: Calibration of a water vapour Raman lidar with a kite-based humidity sensor, Atmos. Meas. Tech., 9, 1083-1094, doi:10.5194/amt-9-1083-2016, 2016. Could you comment on whether this correction of the overlap factor needs to be re-evaluated at the same time as K when the telescope is re-aligned/re-focused? Rather than PMT inhomogeneity, the incidence on the interference filters may have been modified by the change of focus, which is known to have a large impact. In Figure 5, there is a great variability of the observed overlap correction, what can explain this? Lidar noise? Radiosounding error? It may be necessary to distinguish different cases because it is an important point for the robustness of the measurement in the lower tropospheric layers. Can you evaluate or at least comment on the resulting uncertainty on w below 1 km altitude?

Section 3.1.1 Radiosounding errors should also be shown in Figure 6. L24-27. The decrease of the water vapor concentration could be seen on the in-situ measurements of weather stations. Perhaps the temporal evolution of one of these measurements should be added. Why would the laser energy have decreased? Is it because of cold, flash lamps and/or damage on optics? The differences with the modeling can be

related to local effects and thus to the representativity of the measurement site at the mesoscale. They can also be due to a problem in the assimilation process if it does not integrate well the error matrices.

Section 3.1.2 L11. Typing error on "difference-wsonde"?

Section 3.1.3 This section should be merged with section 3.1.1.

Section 3.2 L11. A ground level in-situ measurement could have helped.

Section 3.3.1 It is not so clear whether assimilation is only about radiosounding. Are there no other types of data assimilated, such as spaceborne data? It would be better to show the scatter plot of the radiosounding/LA also.

Section 4 Change numbering.

It's a very good job, congratulations.

───────────────────────────

---

## Referee Comment (RC2) · Anonymous Referee #2 · 23 Jul 2018

The manuscript by Sakai et al. mainly describes the desing and the performances of an automatic Raman lidar system designed for the measurement of water vapor mixing ratio profile during daytime and nighttime conditions. According to the authors, the final goal of the work is to show the positive impact that water vapour Raman lidar measurements may potentially have if assimilated in a heavy-rain forecasting system.

The manuscript is sufficiently well written and outlines in detail the experimental setup of the Raman lidar. The stability of the Raman lidar calibration is assessed over the test period of the instruments, while the correction for the system incomplete overlap is calculated using radiosounding data from a nearby station. Intercomparison statistics versus radiosoundings and GNSS measurements for both the profile and the integrated water vapor content are used to validate the Raman lidar measurements.

[Figure]

Regardless of my specific concerns about the conclusions presented by the authors to support the validation of the Raman lidar measurements, more in general, I think that this manuscript does not demonstrate what the title would like to claim, i.e the positive impact of the Raman lidar measurements on an heavy rain forecasting model.

Focussing on the section where the comparison with high resolution local analysis data is reported, the authors' expectation is to demonstrate, from the Observation-minus-Background (O-B) comparison on a limited time period (less than 5 months), that the Raman lidar can improve the rain forecasting system because it is able to reveal an evident bias in the analysis model output. This is indeed a demonstration of the well known value of Raman lidar measurement to assess the performance of the model analysis output. To demonstrate the impact of lidar observations on any forecasting system a data assimilation experiment or alternatively an Observing System Simulation Experiments (OSSE) must be carried out. Various examples are available in literature of lidar data assimilation experiments (e.g. Wulfmeyer et al., 2006, https://journals.ametsoc.org/doi/10.1175/MWR3070.1). The authors state that they are currently studying the impact of using lidar data with a nonhydrostatic mesoscale model for simulating heavy rainfall in the Kanto area in Summer 2016, citing a paper in Japanese: to my opinion the outcome of these experiment must be embedded in the manuscript by Sakai et al. because it could be the only possibility to add more substance to the manuscript and create a real scientific interest in the readers.

In addition, the lidar described in the manuscript does not add new knowledge about innovative, more advanced technological solutions than the other home-made and commercial Raman lidars operating around the world. Besides, also about the intercomparison of Raman lidar measurements with radiosoundings, GNSS, MWR and FTIR, many other papers are available in literature using more robust approaches (Bhawar et al., 2011, https://rmets.onlinelibrary.wiley.com/doi/pdf/10.1002/qj.697; Beherendt et al., 2007, https://journals.ametsoc.org/doi/full/10.1175/JTECH1924.1).

The authors themselves, when trying to assess of the Raman lidar system performance

which should be able to provide continuous profile of the water vapor mixing ratio, they do clearly show that during daytime the lidar has very limited performance, providing measurements with an uncertainty lower than 30% up to about 1.0-1.5 km above the ground level, which is also the region where the overlap correction is applied. These performances are even lower than a few of commercial Raman lidars and for sure does not allow to achieve the desired impact on a data assimilation system. However, as I said before the impact must be concretely demonstrated and the considerations provided in the manuscript are not sufficient to this purpose. I must also note that the authors honestly acknowledge that the maximum measurement altitude achievable with the Raman lidar system is limited during the daytime and that, though in theory this does not prevent the data assimilation (though I am concerned about the total uncertainty budget in this region), there are the limited information provided by the lidar in the boundary layer and obviously above. This pushes the authors to state that the development of a diode laser-based differential absorption lidar (ongoing) will allow to improve the range and the quality of the measurement for their rain forecasting system. This statement sounds like a "certification" of the insufficient performance of the Raman lidar for the proposed objective.

Therefore, I'd propose the manuscript rejection, but I hope to see the authors submitting soon a new manuscript showing concrete results related to the impact of DIAL measurements or, at least, of the current night time Raman lidar measurements on a rain forecasting system.

---

## Author Comment (AC1) · 29 Aug 2018

**Response to the Reviewer #1**

First of all, we would like to thank the reviewer for their efforts. The comments and questions have been very helpful for improving the manuscript. Please find below the reviewer comments in black, followed by the author's response in blue.

General comment:
The article presented by Sakai et al. is within the topics of AMT. It is clear in its approach and presents a very detailed validation of a new H2O Raman lidar. In addition, the opportunities offered by the water vapor lidars are relevant considering the increase of the extreme raining events. It deserves to be published. I have some minor corrections and remarks listed in the following.

Thank you for reading our manuscript and giving insightful comments and suggestions. We have read your comments carefully and revised the manuscript in accordance with the suggestions. Please find our point-to-point answers below.

1) Abstract L15: Changed their by the Introduction The requirement for data assimilation is more on the absolute value of the root-mean-square-error, less than 0.4 g/kg in the planetary boundary layer (Weckwerth et al). Biases are more problematic for data assimilation process and may induce large discrepancies.

In accordance with the reviewer's suggestion, we have provided the absolute values of the RMSD (0.98 g/kg) as well as the relative error in the abstract, which is larger than the required value of 0.4 g/kg reported by Weckwerth et al. (1999). However, in a recent review by Wulfmeyer et al. (2015) reported that the noise error lower than 10% and bias error smaller 5% are required for the data assimilation. These values can be translated into <1 g/kg and < 0.5 g/kg for 10 g/kg that is typical value in the lower troposphere. Thus, we have provided these percentage values in the Introduction.

2) Section 2.4: The investigation about the variation of K is a very interesting study. From our experience, it may be due to temperature instabilities in the trailer, although in our case we could not pinpoint whether it was due to PMT gain or filter CWL variations. Maybe the temperature of the air conditioning was set differently during the summer and the fall? The high voltages of the photomultipliers seem to be fixed; some authors vary the PMT gains to adjust for the signal/sky background noise ratio during daytime and nighttime. This means the PMT gain has been optimized for daytime limitations, and that the lidar could be more effective at night. Could you comment? Why was it necessary to adjust the focus? Because of the displacement of the trailer? Is your collimating lens an achromat? If not, it could explain the change of K with the change of focus.

We set the temperature of the air conditioner 23°C during the summer and autumn. The variation of the temperature in the trailer was at most ± 5 K (21-28°C). According to the manufacturers, the temperature variation of the sensitivity of PMT is <0.4%/K (Hamamatsu Photonics, Japan) and that of the filter CWL is < 0.0035 nm/K (FUJITOK, Japan), which corresponds to <6% variation of the effective Raman backscattering cross section ratio ($N_2$ / $H_2O$) for ± 5 K variation (Fig. A1). Accordingly, we have added comments on these variations in the revised manuscript.

[Figure]

Figure A1. Calculated differential effective Raman backscattering cross section ratio of $N_2$ to $H_2O$ channels as a function of interference filters used in this study.

As the reviewer has suggested, changing the high voltages of the PMTs during daytime and nighttime is effective to optimize the PMT gain. During the experiment in 2016, we did not change the high voltages (i.e. $-1300$ V) so that the measurement performance was limited. In particular, the sensitivity of PMT decreased around midday in Summer when the solar zenith angle is high. To improve the performance, we upgraded the lidar control program to automatically change the high voltages during day and night in 2018.

3) Section 2.5: To improve the calibration process, especially for the overlap correction function in the lower layers, tethered balloon or kite can be used as in Totems and Chazette (2016). We are then certain of the location of the reference measurements, and we can renew it at will. The accuracy on w is then better. Totems, J. and Chazette, P.: Calibration of a water vapour Raman lidar with a kite-based humidity sensor, Atmos. Meas. Tech., 9, 1083-1094, doi:10.5194/amt-9-1083-2016, 2016. Could you comment on whether this correction of the overlap factor needs to be re-evaluated at the same time as K when the telescope is re-aligned/re-focused? Rather than PMT inhomogeneity, the incidence on the interference filters may have been modified by the change of focus, which is known to have a large impact.

The use of kite for the lidar calibration and determination of the overlap function is a promising method because it can measure the air close to the lidar. One important issue for use of it is that we need to get permission of the Minister of Land, Infrastructure and Transport if we fly it over a height of 150 m in a densely populated area or near airport in Japan. In accordance with the suggestion, we have added the comment on that and referenced Totems and Chazette's paper in the text.
    We agree that the correction of the overlap needs to be re-evaluated when the telescope is re-aligned/refocused. Thank you for the information that change of the telescope's focus has large impact on $K$ by modifying the incidence on the interference filters that.

4) In Figure 5, there is a great variability of the observed overlap correction, what can explain this? Lidar noise? Radiosounding error? It may be necessary to distinguish different cases because it is an important point for the robustness of the measurement in the lower tropospheric layers. Can you evaluate or at least comment on the resulting uncertainty on w below 1 km altitude?

We have checked the individual profile of the comparison of $w$ between the lidar and radiosounding (Fig. S1) . It is difficult for us to distinguish the reasons for the variability. The possible reasons are 1) the difference of the measurement period (i.e. 20 minutes average for the lidar and approximately 1 second for the radiosonding), 2) the difference of the vertical resolution (i.e. 75 m for the lidar and 20– 300 m (it depends on the significant pressure level interval) for the radiosounding data), and 3) lidar noise. We have added these comments on the sources of the variability in the text. In accordance with the reviewer's suggestion, we have added comments on the uncertainty of the correction based on the standard deviation of the difference between the lidar and radiosounding.

5) Section 3.1.1: Radiosounding errors should also be shown in Figure 6.

We have added the error bars of the radiosonding in Fig. 6.

6) L24-27: The decrease of the water vapor concentration could be seen on the in-situ measurements of weather stations. Perhaps the temporal evolution of one of these measurements should be added. Why would the laser energy have decreased? Is it because of cold, flash lamps and/or damage on optics? The differences with the modeling can be related to local effects and thus to the representativity of the measurement site at the mesoscale. They can also be due to a problem in the assimilation process if it does not integrate well the error matrices.

Because the decrease of the water vapor concentration can be seen in Fig. 9, we would like to retain the figure without showing the temporal evolution of $w$ at the surface. Instead, we have added the comment in the text that the monthly mean $w$ values decreased from 17 to 4 g/kg at 1000 hPa and from 8 to 1 g/kg at 700 hPa between August and December in 2016.
     The primary reason for the decrease in the laser power was the aging of the flash lamp because the emission power decreases as increasing shot number (the lifetime is 20 million shots, or about 3 weeks for the continuous operation). In fact, the laser power increased from 110 mJ/pulse to 220 mJ/pulse after replacing the flash lamp and adjusting angles of second and third harmonic crystals after the experiment on 8 December 2016. We have added the comment in the manuscript.
     We agree with you that the differences with the modeling can be related to local effects and thus to the representativeness of the measurement site at the mesoscale. We have added the comment that in the revised manuscript.

7)  Section 3.1.2, L11: Typing error on "difference-wsonde"?

We have deleted the word because it is unnecessary.

8)  Section 3.1.3: This section should be merged with section 3.1.1.

We have merged Section 3.1.3 with Section 3.1.1 as the reviewer's suggestion. To be consistent with this change, we have changed the order of subsections in Section 3.3.

9)  Section 3.2, L11: A ground level in-situ measurement could have helped.

Following the reviewer's suggestion, we used the ground level in-situ measurement of $w$ to compute PWV from the lidar data instead of interpolating the lidar-derived w at 0.14 km to

the ground level. By this change, the lidar-derived PWV values has slightly changed and thus result of the comparison between RL- and GNSS-derived values also changed (Table 3).

10)  Section 3.3.1: It is not so clear whether assimilation is only about radiosounding. Are there no other types of data assimilated, such as spaceborne data? It would be better to show the scatter plot of the radiosounding/LA also.

The LA assimilates the multiple sources, including surface measurements, radiosounding, satellites, and GNSS-derived PWV data, as has been described in Sect. 3 (P5, L15). Because the main purpose of this paper is the validation of the RL system, we would like to show the scatter plot of the radiosounding and LA only in the response (Fig. A2) but not in the original manuscript. We can see in Fig. 2A that there was a negative bias in the LA data.

[Figure]

Fig. A2. (Left panel) Scatter plot of $w$ obtained with the LA ($w_{LA}$) versus $w$ obtained with radiosondes ($w_{Sonde}$) from 2 August to 6 December 2016. (Right panel) Scatter plot of the difference ($w_{LA} - w_{Sonde}$) as a function of $w_{LA}$.

11) Section 4: Change numbering.
Collected.

We wish to thank the reviewer again for his or her valuable comments.

References:

Hamamatsu Photonics K. K. (2007), Photomultiplier tubes, basics and applications, 3rd Ed., Figures 8-11 and 13-1 (available from https://www.hamamatsu.com/resources/pdf/etd/PMT_handbook_v3aE.pdf).
Weckwerth, T. M., V. Wulfmeyer, R. M. Wakimoto, R. M. Hardesty, J. W. Wilson, and R. M. Banta (1999), NCAR-NOAA lower tropospheric water vapor workshop, *Bull. Am. Meteorol. Soc.*, 80, 2339–2357.
Wulfmeyer, V., R. M. Hardesty, D. D. Turner, A. Behrendt, M. P. Cadeddu, P. Di Girolamo, P. Schlüssel, J. Van Baelen, and F. Zus (2015), A review of the remote sensing of lower tropospheric thermodynamic profiles and its indispensable role for the understanding

and the simulation of water and energy cycles, *Rev. Geophys.*, 53, doi:10.1002/2014RG000476.

[Figure]

Figure S1. Vertical distribution of the ratio of *w* obtained by radiosonde (light blue) to *w* obtained with the RL system without beam overlap correction (black) from 2 August to 6 December 2016.

[Figure]

Figure S1. (Contd.)

---

## Author Comment (AC2) · 29 Aug 2018

**Response to the Reviewer #2:**

First of all, we would like to thank the reviewers for the efforts. The comments and suggestions have been very helpful to us. Please find below the reviewer comments in black, followed by author's response in blue.

1) The manuscript by Sakai et al. mainly describes the designing and the performances of an automatic Raman lidar system designed for the measurement of water vapor mixing ratio profile during daytime and nighttime conditions. According to the authors, the final goal of the work is to show the positive impact that water vapour Raman lidar measurements may potentially have if assimilated in a heavy-rain forecasting system.
The manuscript is sufficiently well written and outlines in detail the experimental setup of the Raman lidar. The stability of the Raman lidar calibration is assessed over the test period of the instruments, while the correction for the system incomplete overlap is calculated using radiosounding data from a nearby station. Intercomparison statistics versus radiosoundings and GNSS measurements for both the profile and the integrated water vapor content are used to validate the Raman lidar measurements.
Regardless of my specific concerns about the conclusions presented by the authors to support the validation of the Raman lidar measurements, more in general, I think that this manuscript does not demonstrate what the title would like to claim, i.e. the positive impact of the Raman lidar measurements on an heavy rain forecasting model.

Thank you for the critical comments on our manuscript. We agree with your claim that the manuscript does not show the final goal that is to show the positive impact of the lidar measurements on a heavy rain forecasting model. However, we would like to say that this is the first step of our study aiming to the goal, that is to say, to describe the experimental setup of the low-cost mobile Raman lidar and the validation of measurement by comparisons with other humidity sensors and model. To avoid potential misleading, we would like to change the title to "Mobile water vapor Raman lidar for heavy rain forecasting: instrument description and comparison with radiosonde, GNSS, and high-resolution objective analysis".

The study on the impact of using lidar data on the heavy rain forecast with a nonhydrostatic mesoscale model has been published in English (Yoshida et al. 2018) that showed a positive impact on the analyzed humidity field. We believe that the current manuscript meets the main subject area of AMT that comprise the development, intercomparison, and validation of the measurement instruments.

2) Focusing on the section where the comparison with high resolution local analysis data is reported, the authors' expectation is to demonstrate, from the Observation minus-Background (O-B) comparison on a limited time period (less than 5 months), that the Raman lidar can improve the rain forecasting system because it is able to reveal an evident bias in the analysis model output.
－This is indeed a demonstration of the well known value of Raman lidar measurement to assess the performance of the model analysis output.

We agree with you that the Raman lidar can reveal the bias of the model analysis output. We think that this finding (i.e. positive bias in the local analysis model output from the lidar data) is one important outcome of the study.

3) To demonstrate the impact of lidar observations on any forecasting system a data assimilation experiment or alternatively an Observing System Simulation Experiments (OSSE) must be carried out. Various examples are available in literature of lidar data assimilation experiments (e.g. Wulfmeyer et al., 2006, https://journals.ametsoc.org/doi/10.1175/MWR3070.1). The authors state that they are currently studying the impact of using lidar data with a nonhydrostatic mesoscale model for simulating heavy rainfall in the Kanto area in Summer 2016, citing a paper in Japanese: to my opinion the outcome of these experiment must be embedded in the manuscript by Sakai et al. because it could be the only possibility to add more substance to the manuscript and create a real scientific interest in the readers.

As stated before, we would like to publish separately the result of data assimilation experiment from this manuscript. However, if this manuscript does not meet quality standard of AMT, we should be able to revise the manuscript by embedding the result of data assimilation experiment by Yoshida et al. (2018). In that case, we would like to change the first author to Dr. Yoshida because the main topic of the manuscript would change.

4) In addition, the lidar described in the manuscript does not add new knowledge about innovative, more advanced technological solutions than the other home-made and commercial Raman lidars operating around the world. Besides, also about the intercomparison of Raman lidar measurements with radiosoundings, GNSS, MWR and FTIR, many other papers are available in literature using more robust approaches (Bhawar et al., 2011, https://rmets.onlinelibrary.wiley.com/doi/pdf/10.1002/qj.697; Beherendt et al., 2007, https://journals.ametsoc.org/doi/full/10.1175/JTECH1924.1).

Thank you for introducing the papers on the validation of Raman lidars. By reading those paper, we found that there are more robust approaches of the intercomparison than ours. However, we think that our approach is still useful for the validation because the distances of the measurement instruments were much smaller (less than 100 m) than them.
     In response to your claim that the manuscript does not add new knowledge about innovative or more advanced technological solutions than the existing Raman lidar, we would like to point out that one advanced technological solution of the mobile Raman lidar (MRL) is that it can be easily deployed to remote site and start the measurement in a few hours after the deployment. That is very beneficial for investigating what measurement locations are effective for the heavy rain forecasting. To our knowledge, such a small mobile Raman lidar has only been reported by Chazette et al. (2015) and few intercomparison paper has been available.

5) The authors themselves, when trying to assess of the Raman lidar system performance which should be able to provide continuous profile of the water vapor mixing ratio, they do clearly show that during daytime the lidar has very limited performance, providing measurements with an uncertainty lower than 30% up to about 1.0-1.5 km above the ground level, which is also the region where the overlap correction is applied. These performances are even lower than a few of commercial Raman lidars and for sure does not allow to achieve the desired impact on a data assimilation system.
However, as I said before the impact must be concretely demonstrated and the considerations provided in the manuscript are not sufficient to this purpose.
I must also note that the authors honestly acknowledge that the maximum measurement altitude achievable with the Raman lidar system is limited during the daytime and that, though in theory this does not prevent the data assimilation (though I am concerned about the

total uncertainty budget in this region), there are the limited information provided by the lidar in the boundary layer and obviously above.

Even though the maximum measurement altitude is limited to 1.0-1.5 km in daytime, it is still useful for the heavy rain forecast because the height of the inflow of moist air that can cause heavy rainfall downwind is mostly around 0.5 km in Japan (Kato, 2018). Moreover, Yoshida et al. (2018) has shown a positive impact of the MRL data on the analyzed humidity field as mentioned before. We also note that we intendedly limited the performance of the lidar (but still meets our requirement) to reduce the total cost (< 200 K US dollars) because it makes easier to distribute the MRL around the forecasting areas to increase the chance of catching the inflow.

As for the overlap correction, we estimate the total uncertainty is at most 20% where the overlap correction is applied. We have added this estimation in the revised manuscript.

6) This pushes the authors to state that the development of a diode laser-based differential absorption lidar (ongoing) will allow to improve the range and the quality of the measurement for their rain forecasting system. This statement sounds like a "certification" of the insufficient performance of the Raman lidar for the proposed objective.    Therefore, I'd propose the manuscript rejection, but I hope to see the authors submitting soon a new manuscript showing concrete results related to the impact of DIAL measurements or, at least, of the current night time Raman lidar measurements on a rain forecasting system.

Thank you for encouraging us to further study to improve heavy rain forecasting system. Improving the forecast accuracy and lead time of heavy rain is an urgent issue in Japan. In fact, heavy rain caused floods and landslides that killed over a hundred of people in the southwest Japan in June 2016, July 2017, and July 2018. So we decided to develop the MRL at first before completion of the development of the diode-laser-based DIAL even though the measurement performance was limited. We think that this manuscript is an important step of our study.

We wish to thank the reviewer again for his or her valuable comments.

Sincerely Yours,
Tetsu Sakai

References:

Chazette, P., Marnas, F., and Totems, J.: The mobile water vapor aerosol Raman LIdar and its implication in the framework of the HyMeX and ChArMEx programs: application to a dust transport process, Atmos. Meas. Tech., 7, 1629-1647, https://doi.org/10.5194/amt-7-1629-2014, 2014.
Kato, T., Representative height of the low-level water vapor field for examining the initiation of moist convection leading to heavy rainfall in East Asia. J. Meteor. Soc. Japan, 96, 68-83, 2018.
Yoshida, S., T. Sakai, T. Nagai, S. Yokota, H. Seko, Y. Shoji: Feasibility study of data assimilation using a mobile water vapor Raman lidar, Proceedings of the 19th conference on coherent laser radar technology and applications, p251-255, https://clrccires.colorado.edu/data/paper/P21.pdf, 2018.

---

## Author Response (AR1)

**Author's Responses**

First of all, we would like to thank the reviewers for their efforts. The comments and questions have been very helpful for improving the manuscript. We provided a point-by-point responses to the reviewers' comments, additional changes and corrections, and the marked up manuscript version showing the changes made in red strikethrough and underlines.

**Response to the Reviewer #1**

Please find below the reviewer comments in black, followed by the author's response in blue.

General comment:
The article presented by Sakai et al. is within the topics of AMT. It is clear in its approach and presents a very detailed validation of a new H2O Raman lidar. In addition, the opportunities offered by the water vapor lidars are relevant considering the increase of the extreme raining events. It deserves to be published. I have some minor corrections and remarks listed in the following.

Thank you for reading our manuscript and giving insightful comments and suggestions. We have read your comments carefully and revised the manuscript in accordance with the suggestions. Please find our point-to-point answers below.

1) Abstract L15: Changed their by the Introduction The requirement for data assimilation is more on the absolute value of the root-mean-square-error, less than 0.4 g/kg in the planetary boundary layer (Weckwerth et al). Biases are more problematic for data assimilation process and may induce large discrepancies.

In accordance with the reviewer's suggestion, we have provided the absolute values of the RMSD (0.98 g/kg) as well as the relative error in the abstract, which is larger than the required value of 0.4 g/kg reported by Weckwerth et al. (1999). However, in a recent review by Wulfmeyer et al. (2015) reported that the noise error lower than 10% and bias error smaller 5% are required for the data assimilation. These values can be translated into <1 g/kg and < 0.5 g/kg for 10 g/kg that is typical value in the lower troposphere. Thus, we have provided these percentage values as follows.

P1, L14-16: "The comparison results showed that MRL-derived w agreed within 10% (root-mean-square difference of 0.98 g/kg) with values obtained by radiosonde at altitude ranges".
P2, L1-2: "Wulfmeyer et al. (2015) discussed the requirements of accuracy of the lower tropospheric water vapor measurement for data assimilation and reported that it should be smaller than 10% in noise error and < 5% in bias error."
P16, L26-28: "Our comparison of the MRL-derived w values with those obtained with collocated radiosondes showed that they agreed within 10% and RMSD with 0.98 g/kg between altitudes of 0.14 and 5–6 km at night and between altitudes of 0.14 and 1.5 km in the daytime."

2) Section 2.4: The investigation about the variation of K is a very interesting study. From our experience, it may be due to temperature instabilities in the trailer, although in our case we could not pinpoint whether it was due to PMT gain or filter CWL variations. Maybe the temperature of the air conditioning was set differently during the summer and the fall? The high voltages of the photomultipliers seem to be fixed; some authors vary the PMT gains to

adjust for the signal/sky background noise ratio during daytime and nighttime. This means the PMT gain has been optimized for daytime limitations, and that the lidar could be more effective at night. Could you comment? Why was it necessary to adjust the focus? Because of the displacement of the trailer? Is your collimating lens an achromat? If not, it could explain the change of K with the change of focus.

We set the temperature of the air conditioner 23°C during the summer and autumn. The variation of the temperature in the trailer was at most ± 5 K (21-28°C). According to the manufacturers, the temperature variation of the sensitivity of PMT is <0.4%/K (Hamamatsu Photonics, 2017) and that of the filter CWL is < 0.0035 nm/K (FUJITOK, Japan, personal communication), which corresponds to <6% variation of the effective Raman backscattering cross section ratio ($N_2$ / $H_2O$) for ± 5 K variation (Fig. A1). Accordingly, we have added comments on these variations to Section 2.4 as follows.

P6, L13-16: "During the experimental period, the variation of temperature in the trailer was at most ± 5 K, which corresponds to <6% variation of the effective Raman backscattering cross section ratio and thus $K$, assuming that the temperature variation of the sensitivity of PMT is <0.4%/K (Hamamatsu Photonics, 2017) and that of the filter CWL is < 0.0035 nm/K (FUJITOK, Japan, personal communication)."

As the reviewer has suggested, changing the high voltages of the PMTs during daytime and nighttime is effective to optimize the PMT gain. During the experiment in 2016, we did not change the high voltages (i.e. −1300 V) so that the measurement performance was limited. In particular, the sensitivity of PMT decreased around midday in Summer when the solar zenith angle is high. To improve the performance, we upgraded the lidar control program to automatically change the high voltages during day and night in 2018.

[Figure]

Figure A1. Calculated differential effective Raman backscattering cross section ratio of $N_2$ to $H_2O$ channels as a function of interference filters used in this study.

3) Section 2.5: To improve the calibration process, especially for the overlap correction function in the lower layers, tethered balloon or kite can be used as in Totems and Chazette (2016). We are then certain of the location of the reference measurements, and we can renew it at will. The accuracy on w is then better. Totems, J. and Chazette, P.: Calibration of a water vapour Raman lidar with a kite-based humidity sensor, Atmos. Meas. Tech., 9, 1083-1094, doi:10.5194/amt-9-1083-2016, 2016. Could you comment on whether this correction of the overlap factor needs to be re-evaluated at the same time as K when the telescope is realigned/re-focused? Rather than PMT inhomogeneity, the incidence on the interference filters may have been modified by the change of focus, which is known to have a large impact.

The use of kite for the lidar calibration and determination of the overlap function is a promising method because it can measure the air close to the lidar. We think that unmanned aerial vehicle can be also used for that purpose. One important issue for use of them is that we need to get permission of the Minister of Land, Infrastructure and Transport if we fly it over a height of 150 m in a densely populated area or near airport in Japan. In accordance with the suggestion, we have added the comment with the reference paper (Totems and Chazette, 2016) to Section 2.5 as follows.

P7, L11-13: "The variation should be reduced if using the data measured above the lidar by using a kite (Totems and Chazette, 2016) or unmanned aerial vehicles."

We agree that the correction of the overlap needs to be re-evaluated when the telescope is re-aligned/refocused. Thank you for the information that change of the telescope's focus has large impact on $K$ by modifying the incidence on the interference filters that.

4) In Figure 5, there is a great variability of the observed overlap correction, what can explain this? Lidar noise? Radiosounding error? It may be necessary to distinguish different cases because it is an important point for the robustness of the measurement in the lower tropospheric layers. Can you evaluate or at least comment on the resulting uncertainty on w below 1 km altitude?

We have checked the individual profile of the comparison of $w$ between the lidar and radiosounding (Fig. S1) . It is difficult for us to distinguish the reasons for the variability. The possible reasons are 1) the difference of the measurement period and the temporal resolution (i.e. 20 minutes average for the lidar and approximately 1 second for the radiosonding), 2) the difference of the vertical resolution (i.e. 75 m for the lidar and 20–300 m (it depends on the significant pressure level interval) for the radiosounding data), and 3) lidar noise. The uncertainty of the correction was estimated to be 8% from the standard deviation of the difference between the lidar and radiosounding as follows.

P7, L8-11: 'The uncertainty of the correction was estimated to be 8% from the standard deviation of the profiles. The possible reasons for the variation among the profiles are difference of the measurement period and temporal resolution (i.e. 10 minutes average for the lidar and approximately 1 second for the radiosonde), difference of the vertical resolution (i.e. 75 m for the lidar and 20–300 m that depends on the significant pressure level interval for the radiosonde, and lidar noise."

5) Section 3.1.1: Radiosounding errors should also be shown in Figure 6.

We have added the error bars of the radiosonding in Fig. 6.

6) L24-27: The decrease of the water vapor concentration could be seen on the in-situ measurements of weather stations. Perhaps the temporal evolution of one of these measurements should be added. Why would the laser energy have decreased? Is it because of cold, flash lamps and/or damage on optics? The differences with the modeling can be related to local effects and thus to the representativity of the measurement site at the mesoscale.

They can also be due to a problem in the assimilation process if it does not integrate well the error matrices.

Because the decrease of the water vapor concentration can be seen in Fig. 9, we would like to retain the figure without showing the temporal evolution of $w$ at the surface. Instead, we have added the comment to the Section 3.1.1 that the monthly mean $w$ values decreased from 17 to 4 g/kg at 1000 hPa and from 8 to 1 g/kg at 700 hPa between August and December in 2016 as follows.

P8, L30-32: "As for the water vapor concentration, the monthly mean w values decreased from 17 to 4 g/kg at 1000 hPa and from 8 to 1 g/kg at 700 hPa between August and December in 2016."

The primary reason for the decrease in the laser power was the aging of the flash lamp because the emission power decreases as increasing shot number (the lifetime is 20 million shots, or about 3 weeks for the continuous operation). In fact, the laser power increased from 110 mJ/pulse to 220 mJ/pulse after replacing the flash lamp and adjusting the angles of second and third harmonic crystals after the experiment on 8 December 2016. We have added the comment to Section 3.1.1 as follows.

P8, L29-30: "As for the laser power, it increased from 110 mJ/pulse to 220 mJ/pulse after replacing the flash lamp and adjusting the angles of second and third harmonic crystals on 8 December 2017."

We agree with you that the differences with the modeling can be related to local effects and thus to the representativeness of the measurement site at the mesoscale. We have added the comment to Section 3.3.2 as follows.

P14, L23-25: "The differences with the LA data can be related to local effects and thus to the representativeness of the measurement site at the mesoscale. They can also be due to a problem in the assimilation process if it does not integrate well the error matrices."

7) Section 3.1.2, L11: Typing error on "difference-wsonde"?

We have deleted the word because it is unnecessary.

8) Section 3.1.3: This section should be merged with section 3.1.1.

We have merged Section 3.1.3 with Section 3.1.1 in accordance with the reviewer's suggestion. To be consistent with this change, we have changed the order of subsections in Section 3.3.

9) Section 3.2, L11: A ground level in-situ measurement could have helped.

Following the reviewer's suggestion, we used the ground level in-situ measurement of $w$ to compute PWV from the lidar data instead of interpolating the lidar-derived w at 0.14 km to the ground level. By this change, the lidar-derived PWV values has slightly changed and thus result of the comparison between RL- and GNSS-derived values also changed as follows. The slope and intercept of the regression changed from 0.968 to 0.967 and from −0.229 mm

to −0.142 mm, respectively, and RMSD decreased from 2.88 mm to 2.84 mm. We have updated these values in Table 3. In addition, we have modified the explanation of the method of calculating PWV values as follows.

P12, L15-16: "Below 0.1 km, we interpolated the $w$ data to the ground level in-situ measurement."

10) Section 3.3.1: It is not so clear whether assimilation is only about radiosounding. Are there no other types of data assimilated, such as spaceborne data? It would be better to show the scatter plot of the radiosounding/LA also.

The LA assimilates the multiple sources, including surface measurements, radiosounding, satellites, and GNSS-derived PWV data, as has been described in Section 3 (P7, L27-28). Because the main purpose of this paper is the validation of the RL system, we would like to show the scatter plot of the radiosounding and LA only in Fig. A2 in this response but not in the original manuscript. We can see in Fig. 2A that there was a negative bias in the LA data.

[Figure]

Fig. A2. (Left panel) Scatter plot of $w$ obtained with the LA ($w_{LA}$) versus $w$ obtained with radiosondes ($w_{Sonde}$) from 2 August to 6 December 2016. (Right panel) Scatter plot of the difference ($w_{LA} - w_{Sonde}$) as a function of $w_{LA}$.

11) Section 4: Change numbering.
Collected.

[Figure]

Figure S1. Vertical distribution of the ratio of *w* obtained by radiosonde (light blue) to *w* obtained with the RL system without beam overlap correction (black) from 2 August to 6 December 2016.

[Figure]

Figure S1. (Contd.)

**Response to the Reviewer #2:**

Please find below the reviewer comments in black, followed by author's response in blue.

1) The manuscript by Sakai et al. mainly describes the designing and the performances of an automatic Raman lidar system designed for the measurement of water vapor mixing ratio profile during daytime and nighttime conditions. According to the authors, the final goal of the work is to show the positive impact that water vapour Raman lidar measurements may potentially have if assimilated in a heavy-rain forecasting system.
The manuscript is sufficiently well written and outlines in detail the experimental setup of the Raman lidar. The stability of the Raman lidar calibration is assessed over the test period of the instruments, while the correction for the system incomplete overlap is calculated using radiosounding data from a nearby station. Intercomparison statistics versus radiosoundings and GNSS measurements for both the profile and the integrated water vapor content are used to validate the Raman lidar measurements.
Regardless of my specific concerns about the conclusions presented by the authors to support the validation of the Raman lidar measurements, more in general, I think that this manuscript does not demonstrate what the title would like to claim, i.e. the positive impact of the Raman lidar measurements on an heavy rain forecasting model.

Thank you for the critical comments on our manuscript. We agree with your claim that the manuscript does not show the final goal that is to show the positive impact of the lidar measurements on a heavy rain forecasting model. However, we would like to say that this is the first step of our study aiming to the goal, that is to say, to describe the experimental setup of the low-cost mobile Raman lidar and the validation of measurement by comparisons with other humidity sensors and model. To clarify the purpose of this manuscript and avoid potential misleading, we would like to change the title to "Mobile water vapor Raman lidar for heavy rain forecasting: instrument description and comparison with radiosonde, GNSS, and high-resolution objective analysis".

The study on the impact of using lidar data on the heavy rain forecast with a nonhydrostatic mesoscale model has already been published in English (Yoshida et al. 2018a) that showed a positive impact on the humidity fields that were analyzed and forecasted with the model. More detailed description of the assimilation experiments will be submitted to a peer-reviewed journal soon (Yoshida et al. 2018b). Accordingly, we have changed the former reference in Japanese to these papers in the revised manuscript. Although the current manuscript does not show the positive impact of the Raman lidar measurement on the heavy rain forecasting model, we believe that it meets the main subject area of AMT that comprise the development, intercomparison, and validation of the measurement instruments.

2) Focusing on the section where the comparison with high resolution local analysis data is reported, the authors' expectation is to demonstrate, from the Observation minus-Background (O-B) comparison on a limited time period (less than 5 months), that the Raman lidar can improve the rain forecasting system because it is able to reveal an evident bias in the analysis model output.
– This is indeed a demonstration of the well known value of Raman lidar measurement to assess the performance of the model analysis output.

We agree with you that the Raman lidar can reveal the bias of the model analysis output. We think that this finding (i.e. positive bias in the local analysis model output from the lidar data) is one important outcome of the study. This has been noticed in Section 3.1.1.

3) To demonstrate the impact of lidar observations on any forecasting system a data assimilation experiment or alternatively an Observing System Simulation Experiments (OSSE) must be carried out. Various examples are available in literature of lidar data assimilation experiments (e.g. Wulfmeyer et al., 2006, https://journals.ametsoc.org/doi/10.1175/MWR3070.1). The authors state that they are currently studying the impact of using lidar data with a nonhydrostatic mesoscale model for simulating heavy rainfall in the Kanto area in Summer 2016, citing a paper in Japanese: to my opinion the outcome of these experiment must be embedded in the manuscript by Sakai et al. because it could be the only possibility to add more substance to the manuscript and create a real scientific interest in the readers.

As stated earlier, we would like to publish separately the result of the data assimilation experiment from this manuscript because a substantial amount of description is needed to fully describe the result. To convince the readers that the lidar data has a positive impact on the heavy rainfall forecasting, we added a brief summary of the result of the assimilation experiments by Yoshida et al. (2018a, b) to Section 3.4 as follows.

P16, L14-17: "A first assimilation experiment of the MRL-derived vertical profiles of $w$ into the JMA-NHM using the three-dimensional LETKF for the heavy rainfall forecasting has been reported by Yoshida et al. (2018a), who showed a positive impact on the analyzed and forecast humidity fields on the Kanto Plain on 17 August 2016. More detailed description of the assimilation experiment will follow soon (Yoshida et al. 2018b)."

4) In addition, the lidar described in the manuscript does not add new knowledge about innovative, more advanced technological solutions than the other home-made and commercial Raman lidars operating around the world. Besides, also about the intercomparison of Raman lidar measurements with radiosoundings, GNSS, MWR and FTIR, many other papers are available in literature using more robust approaches (Bhawar et al., 2011, https://rmets.onlinelibrary.wiley.com/doi/pdf/10.1002/qj.697; Beherendt et al., 2007, https://journals.ametsoc.org/doi/full/10.1175/JTECH1924.1).

Thank you for introducing the papers on the validation of Raman lidars. By reading those paper, we found that there are more robust approaches of the intercomparison than ours. However, we think that our approach is still useful for the validation because the distances of the measurement instruments were much smaller (less than 100 m) than them.
    In response to your claim that the manuscript does not add new knowledge about innovative or more advanced technological solutions than the existing Raman lidar, we would like to point out that one advanced technological solution of the MRL is that it can be easily deployed to remote site and start the measurement in a few hours after the deployment. That is very beneficial for investigating measurement locations that are effective for the heavy rain forecasting. To our knowledge, such a small mobile Raman lidar has only been reported by Chazette et al. (2015) and few intercomparison paper has been available. To emphasize the new aspects of our developments and studies, we have added these comments to the abstract and the Introduction section of the revised manuscript as follows.

P1, L11-13: "The MRL was installed in a small trailer for easy deployment to the upwind side of potential rainfall areas and can start measurement in a few hours to monitor the inflow of moist air before rainfall events."

P1, L29-30: "The MRL can be easily deployed at a site upwind of a potential heavy 30 rainfall area and start measurement in a few hours to monitor the vertical water vapor distribution before a rainfall event."
P2, L10: "To our knowledge, there are few reports on the validation for such a compact mobile system."

5) The authors themselves, when trying to assess of the Raman lidar system performance which should be able to provide continuous profile of the water vapor mixing ratio, they do clearly show that during daytime the lidar has very limited performance, providing measurements with an uncertainty lower than 30% up to about 1.0-1.5 km above the ground level, which is also the region where the overlap correction is applied. These performances are even lower than a few of commercial Raman lidars and for sure does not allow to achieve the desired impact on a data assimilation system.
However, as I said before the impact must be concretely demonstrated and the considerations provided in the manuscript are not sufficient to this purpose.
I must also note that the authors honestly acknowledge that the maximum measurement altitude achievable with the Raman lidar system is limited during the daytime and that, though in theory this does not prevent the data assimilation (though I am concerned about the total uncertainty budget in this region), there are the limited information provided by the lidar in the boundary layer and obviously above.

Even though the maximum measurement altitude is limited to 1.0-1.5 km in daytime, it is still useful for the heavy rain forecast because the height of the inflow of moist air that can cause heavy rainfall downwind is mostly around 0.5 km in Japan (Kato, 2018). Moreover, Yoshida et al. (2018a) has shown a positive impact of the MRL data on the analyzed and forecast humidity fields as mentioned before. We also note that we intendedly limited the performance of the lidar (but still meets our requirement) to reduce the total material cost (< 250K US dollars) because it makes easier to distribute the MRL around the forecasting areas to increase the opportunity of detecting inflow. We have added this comment to the Introduction section of the revised manuscript as follows.

P2, L3-5: "Besides the requirement of measurement accuracy, reducing the cost of the lidar is important because it makes easier to distribute them around the forecasting area to increase the opportunity of detecting the inflow. We developed our mobile MRL system to meet these requirements as much as possible within the total material cost of ~250K USD."

As for the overlap correction, we estimate that the total uncertainty is at most 23% (10% for the calibration coefficient, 8% for the overlap correction, and <5% for the signal noise) where the overlap correction is applied. We have provided the estimated values of the uncertainty of the overlap correction in Section 2.5 as follows.

P7, L8: "The uncertainty of the correction was estimated to be 8% from the standard deviation of the profiles."

6) This pushes the authors to state that the development of a diode laser-based differential absorption lidar (ongoing) will allow to improve the range and the quality of the measurement for their rain forecasting system. This statement sounds like a "certification" of the insufficient performance of the Raman lidar for the proposed objective. Therefore, I'd propose the manuscript rejection, but I hope to see the authors submitting soon a new

manuscript showing concrete results related to the impact of DIAL measurements or, at least, of the current night time Raman lidar measurements on a rain forecasting system.

Thank you for encouraging us to further study to improve heavy rain forecasting system. Improving the forecast accuracy and lead time of heavy rain is an urgent issue in Japan. In fact, heavy rain caused floods and landslides that killed over a hundred of people in the southwest Japan in June 2016, July 2017, and July 2018. So we decided to develop the MRL at first before completion of the development of the diode-laser-based DIAL even though the measurement performance was limited. We think that this manuscript is an important step of our study.

[revised manuscript text omitted]

---

## Author Response (AR2)

**Response to Report #3 (Reviewer #3)**

Thank you for reviewing our manuscript. Our response to the reviewer's comments and the corresponding text from the manuscript are presented below in blue and green fonts, respectively.

**Reviewer's comment:** The paper by Sakai et al. describes the setup of a trailer with a UV Raman lidar for water vapor measurements. This paper is a useful reference for future research with this system's data and data assimilation. The authors want to use the system for heavy rain forecasting. However, I still agree with reviewer two. This paper is not about heavy rain forecasting with a Raman lidar, as the title still suggests. And therefore I would still recommend to drop this part from the title. Maybe you can find an acronym for the lidar system name, that includes ". for heavy rain forecasting". And that acronym/name can be mentioned in the title. Overall, I would recommend the paper for publication, but I have some minor comments:

**Our response:**
From the Reviewers' comments, we realized that the real scope of this paper is to introduce the new compact mobile Raman lidar system, not about the heavy rain forecasting. Therefore, in accordance with the Reviewer's suggestions, we have changed the title of the manuscript from:

"Mobile water vapor Raman lidar for heavy rain forecasting: system instrument description and validation by comparison with radiosonde, GNSS, and high-resolution objective analysis"

to

"Automated compact mobile Raman lidar for water vapor measurement: instrument description and validation by comparison with radiosonde, GNSS, and high-resolution objective analysis".

Please find our point-to-point response to the Reviewer's comments below.

**Reviewer comment:**
1) The general problem, when the data shall be used for short-time forecasts of heavy rain events, is probably the occurrence of clouds. Whenever the water-vapor content is critically large, there are also low clouds, which in turn prohibit optimal lidar measurements. Therefore many people use synergistic approaches with different measurement systems (such as lidar, microwave radiometer, radar, GPS, ...) for such a task. This fact should be mentioned. Lidar is an important tool but not the only required instrument here.

**Our response:**
I agree with the Reviewer's comments that the lidar is unable to measure the water vapor distribution above the optically-thick clouds and thus it is important to use synergistic approaches with different measurement systems to measure the spatial and temporal distribution of the water vapor regardless of cloud occurrence. To mention this, we have added the following sentence (P16, L17-20):

"Despite the potential usefulness of the MRL-measured data for the weather forecasting, the MRL cannot measure the water vapor inside and above optically thick clouds. To overcome this disadvantage, it is important to use synergistic approaches with different instruments

such as GNSS, microwave radiometer, radiosonde to measure the water vapor distribution even under cloudy conditions (e.g., Foth and Pospichal, 2017)."

**Reviewer comment:**
2) Table 1: This table could be skipped and the few requirements written in one sentence. There are also no proper citations for these requirements, where do they come from?
**Our response:**
These requirements came from our discussion with the scientists involved in model development and implementation in our Institute. In accordance with the Reviewer's suggestion, we have deleted Table 1 and modified the sentences on the requirement in the Introduction as following (P1, L34-36):

"Given the temporal and vertical resolutions of the model and the assimilation window length, the required measurement resolutions are at most 30 min in time and 200 m in vertical."

**Reviewer comment:**
3) Table 2: The given beam divergence is before or after beam expansion? Please specify.
**Our response:**
It is after the beam expander. To specify this, we have added a notation in Table 2.

**Reviewer comment:**
4) Figure 1: The two lines from Laser and PIN to the electronics, should be drawn below the receiver box. Right now one line is on top of the beam path which is a bit confusing.
**Our response:**
In accordance with the Reviewer's comment, we have moved the two lines from the laser and PIN to the electronics below the receiver box. In addition to this change, we have added the window and light baffle in Fig. 1 in response to the reviewer's comment 7.

**Reviewer comment:**
5) P4L2: "The temperature is maintained to 22-32 degree C." That sounds not very stable. Can you find a few words on the actual stability? It seems that with a field of view of 0.3 mrad temperature stability, and thus laser beam alignment, is very crucial. Do you have any details on that, how the temperature affects the overlap alignment? Do you have a camera to check the alignment?
**Our response:**
The range of the variability of the temperature (22 and 32 degree Celsius) is the minimum and maximum values for the 5 months during the experimental period. The variation of air temperature in a day was mostly within a range of ±2 °C. We did not find any change in laser and the overlap alignments. To avoid confusion of the readers, we have modified the text as following (P3, L7-9):

"The temperature inside the trailer is maintained within a range of ±2°C between 22 °C and 32 °C by an air conditioner. We did not find any change of the optical alignment of the transmitter and receiver with the change of the temperature."

**Reviewer comment:**
6) How is the fused silica window constructed? Do you transmit and receive through the same window? Are there problems (especially if the window is dirty) with detector overloading from the scattering of the laser beam?

**Our response:**
It was constructed by the Kiyohara Optics Inc. ([http://www.koptic.co.jp/opt/eng/index.html](http://www.koptic.co.jp/opt/eng/index.html)). We have added the manufacture's name in the text (P3, L10). We transmitted and received the light through the same window. There have not been any problems with detector overloading from the scattering of the laser beam. This might be partly owing to the light baffle between the window and telescope (Please see the modified version of Fig.1) that prevented light scattered by the window from entering the telescope. The other reason for that is rainwater occasionally washed away the dust on the window.

**Reviewer comment:**
7) It is mentioned that the data acquisition is analog and photon counting. Do you have any comments on the gluing procedure, or do you only use p.c. data, do you want to present any figure that shows the raw signals?

**Our response:**
We glued the analog and photon counting data. To clarify, we have added a following sentence (P4, L5-7):

"The $P_X$ (z) for each receiving channel was obtained by connecting the photon counting and analog data using a count rate range of 1–10 MHz (mostly 0.2–0.4 km for the water vapor and 0.5–0.9 km for the nitrogen channels) to gain high dynamic range."

**Reviewer comment:**
8) Fig 10, P8L39. A very narrow interference filter is used. Did you consider a temperature correction of the 407 nm signal with help of the filter's transmission curve? This might be the reason for the discrepancies above 5 km height. At least mention that the issue might be relevant here.

**Our response:**
We have evaluated the temperature variation of the effective Raman cross section (i.e. integrated Raman backscattering cross section multiplied by filter's transmission) using the filter's transmission curves provided from the manufacture and the Raman cross sections (Penny and Lapp, 1976; Inaba, 1976; Avila et al., 2014). We found that the variation is 0.5% for the temperature range of 253–303 K (Fig. A1). Thus, the issue is negligible for the reason for the discrepancy. We have added a following comment (P9, L1-3):

"The influence of the temperature dependence of the Raman cross section (e.g. Whiteman, 2003) is negligible for the MRL because the variation is estimated to be 0.5% for the temperature range of 253–303 K."

[Figure]

Figure A1. Temperature variation of difference of the ratio of effective Raman cross section of nitrogen to water vapor channels of MRL.

**Reviewer comment:**
9) Fig 11/13/14/ Table 3: When there is the calculation of the slope and intercept, I suggest to also give the statistical error of these values from the fit. Then you can determine if the slope difference from 1/unity is indeed significant or not.
**Our response:**
In accordance with the Reviewer's comment, we have added the statistical uncertainties of the slopes and intercepts of the regression analysis in the revised manuscript of Table 2 and the text (P11, L14-15). We note that the results of the regression analysis have been slightly changed because we have corrected the number of the data points by deleting the points that were mistakenly counted before.

**Reviewer comment:**
10) P14L17. The first sentence misses a dot.
**Our response:** Corrected.

**Reviewer comment:**
11) P17L3-5: This is a single sentence paragraph. I suggest connecting the sentence to the previous paragraph.
**Our response:** Corrected.

**Reviewer comment:**
12) I suggest harmonizing the figures for equal font size, line width, and resolution.
**Our response:**
In accordance with the suggestion, we have harmonized the figures for font size, line width and resolution as possible as we could.

**Reviewer comment:**
13) I realized that there are not many references within this manuscript, although many have been published on the topic of water vapor Raman lidars, calibration, and assimilation. Here are just a few that might be useful:

1. Dai, G., Althausen, D., Hofer, J., Engelmann, R., Seifert, P., Bühl, J., Mamouri, R.-E., Wu, S., and Ansmann, A.: Calibration of Raman lidar water vapor profiles by means of AERONET photometer observations and GDAS meteorological data, Atmos. Meas. Tech., 11, 2735-2748, https://doi.org/10.5194/amt-11-2735-2018, 2018.

2. Foth, A. and Pospichal, B.: Optimal estimation of water vapour profiles using a combination of Raman lidar and microwave radiometer, Atmos. Meas. Tech., 10, 3325-3344, https://doi.org/10.5194/amt-10-3325-2017, 2017.

3. Foth, A., Baars, H., Di Girolamo, P., and Pospichal, B.: Water vapour profiles from Raman lidar automatically calibrated by microwave radiometer data during HOPE, Atmos. Chem. Phys., 15, 7753-7763, https://doi.org/10.5194/acp-15-7753-2015, 2015.

4. Bhawar et. all 2011, The water vapour intercomparison effort in the framework of the Convective and Orographically induced Precipitation Study: airborne to ground based and airborne to airborne lidar systems

https://doi.org/10.1002/qj.697

5. Herold, C., D. Althausen, D. Müller, M. Tesche, P. Seifert, R. Engelmann, C. Flamant, R. Bhawar, and P. Di Girolamo, 2011: Comparison of Raman Lidar Observations of Water Vapor with COSMO-DE Forecasts during COPS 2007. Wea. Forecasting, 26, 1056–1066, https://doi.org/10.1175/2011WAF2222448.1

6. Grzeschik, Matthias & Bauer, Hans-Stefan & Wulfmeyer, Volker & Engelbart, Dirk & Wandinger, Ulla & Mattis, Ina & Althausen, Dietrich & Engelmann, Ronny & Tesche, Matthias & Riede, Andrea. (2008). Four-Dimensional Variational Data Analysis of Water Vapor Raman Lidar Data and Their Impact on Mesoscale Forecasts. Journal of Atmospheric and Oceanic Technology - J ATMOS OCEAN TECHNOL. 25. 10.1175/2007JTECHA974.1.

**Our response:**
Thank you for introducing the references that are useful for readers. We have added them in the manuscript (i.e. Ref. 1 in P5, L35; Ref. 2 in P16, L20; Refs. 4 and 5 in P7, L22, and Ref. 6 in P1, L32). In addition to these references, we have added the following references that are relevant for the topic of the manuscript.

On the validation of the Raman lidar (P7, L22):
Behrendt, A., V. Wulfmeyer, H. Bauer, T. Schaberl, P. Di Girolamo, D. Summa, C. Kiemle, G. Ehret, D.N. Whiteman, B.B. Demoz, E.V. Browell, S. Ismail, R. Ferrare, S. Kooi, and J. Wang: Intercomparison of Water Vapor Data Measured with Lidar during IHOP_2002. Part I: Airborne to Ground-Based Lidar Systems and Comparisons with Chilled-Mirror Hygrometer Radiosondes, J. Atmos. Oceanic Technol., 24, 3–21, https://doi.org/10.1175/JTECH1924.1, 2007.

On the data assimilation of the lidar data (P1, L32):
Bielli, S., Grzeschik, M. , Richard, E. , Flamant, C. , Champollion, C. , Kiemle, C. , Dorninger, M. and Brousseau, P.: Assimilation of water   vapour airborne lidar observations: impact study on the COPS precipitation forecasts, Q. J. R. Meteorol. Soc., 138: 1652-1667. doi:10.1002/qj.1864, 2012.

Wulfmeyer, V., H. Bauer, M. Grzeschik, A. Behrendt, F. Vandenberghe, E.V. Browell, S. Ismail, and R.A. Ferrare: Four-Dimensional Variational Assimilation of Water Vapor Differential Absorption Lidar Data: The First Case Study within IHOP_2002, Mon. Wea. Rev., 134, 209–230, https://doi.org/10.1175/MWR3070.1, 2006.

On the calibration of water vapor mixing ratio (P5, L35):
David, L., Bock, O., Thom, C., Bosser, P., and Pelon, J.: Study and mitigation of calibration factor instabilities in a water vapor Raman lidar, Atmos. Meas. Tech., 10, 2745-2758, https://doi.org/10.5194/amt-10-2745-2017, 2017.

In addition to these changes, we have added the following sentence in the Conclusion to clarify that the MRL can also be utilized for the study of water vapor in the lower troposphere (P19, L5-7):
"Although the MRL system was originally developed for heavy rain forecasting, it can also be utilized for the study of water vapor in the lower troposphere such as boundary layer structure and cloud formation."

We mention that the sentences explaining the specification of the radiosonde and the local analysis data originally in Sec. 3 have been moved to Sec. 3.1 and 3.3, respectively, to be easy to read.

Thank you again for his or her valuable comments.

In the paper by Yoshida et al., the results collected over about two months-experiment are condensed in one example where there is a certain impact due to the Raman lidar data assimilation though this is not striking when the heavier rain fall comes. Nevertheless, it's not my role to judge the other paper but I think that for sure the paper does not show a large benefit in assimilating Raman lidar water vapor measurements up to 1.0-1.5 km during daytime. Though this may be theoretically useful, it must be demonstrated that the measurements of their system, close and within the incomplete overlap region are really useful and cannot impact negatively the forecasts.

To this purpose, the authors provide an estimation of the total uncertainty for the overlap correction of at most 23%, which is quite high in a region where the atmospheric variability requires a higher accuracy for the data to have an impact on the forecast model.

My opinion is that there is still limited interest in having another manuscript which mainly does not provide any special advance in the usage of the Raman lidar technique for measuring water vapour. The interest in the readers can only come from the fact that a new compact Raman lidar system for monitoring water vapour is available (also on the market?) and can be operated in a measurement network or at a remote station. The manuscript does not need to necessarily be linked to the scope of improving rainfall forecasting but to many other applications, like for any other Raman lidar system. The manuscript does not reveal any special feature of this lidar system which can significantly improve the weather forecasts with respect to other systems previously presented in literature.

Therefore, the manuscript must be necessarily reshaped to consider the real scope of this paper: to introduce another automated compact Raman lidar for water vapour measurements, which has an affordable cost (<250K US dollars) and apparently easy to be operated. I do not think this requires a strong effort but the manuscript will benefit from this work and may be of some interest to the community.

Under these premises, if the editor feels that this is line with the scope of the journal, I recommend major revisions and I am happy to further support the Editor in any following step of this review procedure.

**Response:**

We agree with the Reviewers' comment and realized that the real scope of this paper is to introduce the automated compact Raman lidar for water vapour measurements which has an affordable cost and easy to be operated, not necessarily to improve rainfall forecasting. Therefore, we have changed the title from:

"Mobile water vapor Raman lidar for heavy rain forecasting: system instrument description and validation by comparison with radiosonde, GNSS, and high-resolution objective analysis"

to

"Automated compact mobile Raman lidar for water vapor measurement: instrument description and validation by comparison with radiosonde, GNSS, and high-resolution objective analysis".

Further, to clarify the scope of the paper we have modified the Abstract and Conclusion. For example, we have changed the first sentence of the Abstract from (P1, L9-10):

"To improve the lead time and accuracy of predictions of localized heavy rainfall, which can cause extensive damage in urban areas in Japan, we developed a mobile Raman lidar (MRL) system for measuring the vertical distribution of the water vapor mixing ratio (w) in the lower troposphere."

to

"We developed an automated compact mobile Raman lidar (MRL) system for measuring the vertical distribution of the water vapor mixing ratio ($w$) in the lower troposphere which has an affordable cost and easy to be operated."

We have also changed the last sentence of the Abstract from (P1, L19-20):

"Four months of continuous operation of the MRL system demonstrated its utility for monitoring water vapor distributions for heavy rain forecasting."

to

"Four months of continuous operation of the MRL system demonstrated its utility for monitoring water vapor distributions in the lower troposphere."

Similarly, we have changed the first sentence in the Conclusion from (P16, L26-28):

"We developed a mobile Raman lidar system for measuring the vertical distribution of the water vapor mixing ratio w in the lower troposphere to improve the accuracy and lead time of heavy rainfall prediction."

to

"We developed a low-cost, automated compact mobile Raman lidar system for measuring the vertical distribution of the water vapor mixing ratio w in the lower troposphere, which is easy to be deployed to remote sites and is capable of unattended operation for several months."

And we have added the following sentence to clarify that the MRL can be utilized not only for rainfall forecasting but also the other study of the water vapor in the lower troposphere (P17, L5-6):

"Although the MRL system was originally developed for heavy rain forecasting, it can also be utilized for the study of water vapor in the lower troposphere."

In addition to these changes, we have revised the manuscript in accordance with the comments from the Reviewer #3. Please find the marked-up manuscript version showing the

changes we have made. We trust that the revised manuscript is in line with the scope of the journal.

Thank you again for his or her valuable comments.

[revised manuscript text omitted]